# Structural insights into lipid membrane binding by human ferlins

Constantin Cretu [ID] [1,2,3 ✉], Aleksandar Chernev[4], Csaba Zoltán Kibédi Szabó[5], Vladimir Pena[5], Henning Urlaub[3,4,6], Tobias Moser [ID] [1,3,7 ✉] & Julia Preobraschenski [ID] [1,2,3 ✉]

## Abstract

**Ferlins are ancient membrane proteins with a unique architecture, and play central roles in crucial processes that involve Ca²⁺-dependent vesicle fusion. Despite their links to multiple human diseases and numerous functional studies, a mechanistic understanding of how these multi-C$_2$ domain-containing proteins interact with lipid membranes to promote membrane remodelling and fusion is currently lacking. Here we obtain near-complete cryo-electron microscopy structures of human myoferlin and dysferlin in their Ca²⁺- and lipid-bound states. We show that ferlins adopt compact, ring-like tertiary structures upon membrane binding. The top arch of the ferlin ring, composed of the C$_2$C-C$_2$D region, is rigid and exhibits only little variability across the observed functional states. In contrast, the N-terminal C$_2$B and the C-terminal C$_2$F-C$_2$G domains cycle between alternative conformations and, in response to Ca²⁺, close the ferlin ring, promoting tight interaction with the target membrane. Probing key domain interfaces validates the observed architecture, and informs a model of how ferlins engage lipid bilayers in a Ca²⁺-dependent manner. This work reveals the general principles of human ferlin structures and provides a framework for future analyses of ferlin-dependent cellular functions and disease mechanisms.**

**Keywords** Ferlins; C$_2$ Domain; Cryo-EM; Membrane Fusion; Ca²⁺ Sensing and Signalling
**Subject Categories** Membranes & Trafficking; Structural Biology

## Introduction

Excitable cells rely on precisely timed Ca²⁺ signals to trigger exocytosis in neural synapses and contraction in muscle cells (Clapham, 2007; Jahn and Fasshauer, 2012; Kuo and Ehrlich, 2015; Luan and Wang, 2021). Less is known how the uncontrolled influx of Ca²⁺ through a large membrane lesion promotes a rapid acute response (Cooper and Head, 2015; Cooper and McNeil, 2015; Lek et al, 2012). C$_2$-domain proteins are key molecular players utilizing the incoming Ca²⁺ signal to tether intracellular vesicles to their target membranes and ultimately promote their fusion by coordinated binding to phospholipids and other factors (Andrews and Chakrabarti, 2005; Rizo, 2022; Rizo and Sudhof, 1998). Beyond the well-studied synaptotagmins, which feature only two consecutive C$_2$ domains (Brunger and Leitz, 2023; Jahn et al, 2024; Rizo and Sudhof, 1998; Sudhof, 2013), ferlins, predicted to comprise up to eight C$_2$ domains (Dominguez et al, 2022), play a pivotal role in mediating these processes and are critically needed at multiple pathway steps (Bansal and Campbell, 2004; Cooper and Head, 2015; Cooper and McNeil, 2015; Pangrsic et al, 2012).

Ferlins, such as dysferlin (FER1L1), myoferlin (FER1L3), and otoferlin (FER1L2), form an ancient group of C$_2$ domain Ca²⁺-sensing proteins present in almost all eukaryotic lineages (Bansal and Campbell, 2004; Han and Campbell, 2007; Pangrsic et al, 2012; Petit et al, 2023). Dysferlin and myoferlin are highly expressed in skeletal and heart muscle cells, which are prone to membrane injuries during contractions (Bansal and Campbell, 2004; Cooper and McNeil, 2015; Pramono et al, 2009). Distributed at the sarcolemma, its specialized internal structures and various endomembrane vesicles (Paulke et al, 2024), dysferlin has been proposed to play major roles in Ca²⁺-dependent membrane resealing, in the biogenesis and maintenance of the transverse-tubules system (Kerr et al, 2013; Paulke et al, 2024), and other trafficking pathways (Cooper and McNeil, 2015; Glover and Brown, 2007; Han and Campbell, 2007). The central physiological role of dysferlin in maintaining the integrity of muscle cell membranes is evident from over 400 disease-causing *DYSF* mutations identified in limb-girdle muscle dystrophy type 2B (LGMD2B) and Miyoshi myopathy (MM)—rare autosomal recessive muscle wasting disorders characterized by deficiencies in sarcolemma repair (Bansal et al, 2003; Cooper and Head, 2015). Similar to dysferlin, myoferlin (Cooper and McNeil, 2015; Davis et al, 2000; de Morree et al, 2010), has been linked to diverse membrane remodelling and organelle repair events, including in other cell types, as well as to myoblast membrane fusion during myogenesis (Davis et al, 2000;

[1]Institute for Auditory Neuroscience and InnerEarLab, University Medical Center Göttingen, Göttingen, Germany. [2]Biochemistry of Membrane Dynamics Group, Institute for Auditory Neuroscience, University Medical Center Göttingen, Göttingen, Germany. [3]Cluster of Excellence "Multiscale Bioimaging: from Molecular Machines to Networks of Excitable Cells" (MBExC), University of Göttingen, Göttingen, Germany. [4]Bioanalytical Mass Spectrometry Group, Max Planck Institute for Multidisciplinary Sciences, Göttingen, Germany. [5]Research Group Mechanisms and Regulation of Splicing, The Institute of Cancer Research, London, UK. [6]Bioanalytics Group, Institute for Clinical Chemistry, University Medical Center Göttingen, Göttingen, Germany. [7]Auditory Neuroscience and Synaptic Nanophysiology Group, Max-Planck-Institute for Multidisciplinary Sciences, Göttingen, Germany. ✉E-mail: constantin.cretu@med.uni-goettingen.de; tmoser@gwdg.de; julia.preobraschenski@med.uni-goettingen.de

Doherty et al, 2005). However, unlike dysferlin, myoferlin has recently also been found to be overexpressed in various human cancers and the altered vesicular trafficking and function of myoferlin has been linked to cancer cell proliferation, metastasis, and resistance to chemotherapy (Cooper and McNeil, 2015; Gupta et al, 2021; Zhang et al, 2018). The closely related otoferlin is mainly expressed in sensory hair cells of the inner ear, and several hundred pathogenic mutations cause the deafness DFNB9 (Moser and Starr, 2016; Petit et al, 2023; Santarelli et al, 2015; Vona et al, 2020; Yasunaga et al, 1999). In mechanistic terms, it has been suggested that otoferlin is involved in $Ca^{2+}$-sensing for synaptic vesicle fusion (Johnson and Chapman, 2010; Michalski et al, 2017; Roux et al, 2006) and replenishment (Pangrsic et al, 2010; Vogl et al, 2016), as well as exocytosis-endocytosis coupling (Duncker et al, 2013; Jung et al, 2015; Kroll et al, 2019; Strenzke et al, 2016), and its mutations disrupt synaptic sound encoding, resulting in an auditory synaptopathy (Moser and Starr, 2016; Pangrsic et al, 2012). Despite the recent insights into the individual roles of ferlins and them representing targets for gene therapy or drug discovery (Al-Moyed et al, 2019; Gupta et al, 2021; Llanga et al, 2017; Moser et al, 2024; Zhang et al, 2018), the underlying molecular mechanisms, especially concerning how they promote membrane remodelling and fusion through concerted $Ca^{2+}$ and phospholipid binding, have remained largely unresolved, in part, due to limited structural information on their active states.

In structural terms, ferlins arguably possess the most unique and complex architecture among the known $Ca^{2+}$-sensitive $C_2$-domain factors (Lek et al, 2012; Pangrsic et al, 2012). The ferlin amino-terminal (N-terminal) cytoplasmic domain is predicted to contain up to eight β-sandwich $C_2$ domains (Dominguez et al, 2022). The ferlin $C_2$ domains are thought to be connected by unstructured linker regions and are followed by a carboxy-terminal (C-terminal), single-pass transmembrane region, anchoring the proteins to cellular membranes. In addition, type-I ferlins, such as dysferlin and myoferlin, have two accessory domains with an unknown function, DysF and FerA, inserted between the third and fourth $C_2$ domains (Dominguez et al, 2022). Generally, similar to synaptotagmins (Chapman, 2008; Corbalan-Garcia and Gomez-Fernandez, 2014; Rizo, 2022; Rizo and Sudhof, 1998), the individual $C_2$ domains of ferlins have variable $Ca^{2+}$ and phospholipid binding activities (Abdullah et al, 2014; Marty et al, 2013), with few exceptions (otoferlin's $C_2A$ domain (Helfmann et al, 2011)). However, previous studies failed to clarify how full-length ferlins are precisely organized to interact with lipid membranes and which of their structural motifs are critical for membrane binding. Consequently, it remains poorly understood how ferlins act on lipid membranes to promote their remodelling and fusion, and how $Ca^{2+}$-sensitive conformational rearrangements mediate these processes (Lek et al, 2012; Pangrsic et al, 2012; Xu et al, 2011).

Herein, we leverage structural biology approaches and functional analyses to obtain near complete cryo-EM models of the two largest human ferlins, myoferlin and dysferlin, in their $Ca^{2+}$ and lipid-bound states. Besides revealing the intricate organization of these essential vesicle trafficking factors, our ferlin structures shed light on how the $C_2$ and accessory motifs engage lipid bilayers in a coordinated manner. We further advance a model of how these ferlins cycle between alternative conformational states to transiently bind lipid membranes and facilitate their remodelling and fusion.

# Results

## 2.4 Å cryo-EM structure of membrane-bound human myoferlin

To obtain the first complete structure of a human ferlin, we established the heterologous expression and purification of full-length myoferlin and dysferlin (Fig. 1A; Appendix Fig. S1A). In addition to the membrane-anchored constructs, we expressed the entire cytosolic region of the ferlins, comprising all $C_2$ and accessory motifs (Appendix Fig. S1A). Generally, the soluble, detergent or liposome reconstituted samples were homogeneous and retained their ability to bind $Ca^{2+}$ and negatively charged lipid membranes, as previously reported for the individual ferlin domains (Abdullah et al, 2014; Marty et al, 2013; Padmanarayana et al, 2014) (Appendix Fig. S1B–G). However, in contrast to an early model of dysferlin (Xu et al, 2011) and a recent report (Huang et al, 2024), we did not observe a significant tendency of the proteins to dimerize through the $C_2$ domains, in mass photometry measurements and size-exclusion chromatography, suggestive of ferlins being organized as a monomers in solution (Appendix Fig. S1B,C,H).

Initial attempts at single-particle cryo-EM imaging of lipid-free ferlins revealed the intrinsic flexibility of the N-terminal ($C_2A$-$C_2B$) and C-terminal domains ($C_2F$-$C_2G$), which limited the resolution of the maps. We, therefore, hypothesized that these more dynamic motifs, all predicted to engage lipid bilayers (Abdullah et al, 2014; Johnson and Chapman, 2010; Kwok et al, 2023; Marty et al, 2013), would become organized upon membrane binding. In our efforts to identify an optimal membrane system, we observed that the cytosolic region of myoferlin (residues 1–1997) formed stable complexes with large MSP2N2 (membrane scaffold protein 2 N2)-based lipid nanodiscs, comprising anionic phospholipids and chosen to accommodate all interacting domains (Cannon et al, 2023) (Fig. 1B–D; Appendix Fig. S1B,C). We further stabilized the myoferlin-nanodisc complexes through glutaraldehyde cross-linking (Kastner et al, 2008) and imaged four different protein-lipid samples, which were assembled on membranes containing PS (Phosphatidylserine) alone or combined PS and PI(4,5)$P_2$ (Phosphatidylinositol-4,5-bisphosphate) (Appendix Figs. S2–S5). Computational image sorting allowed us to identify intact particles and reconstruct near-complete cryo-EM maps of the lipid-bound myoferlin (Fig. EV1; Appendix Figs. S2–S5, and Movie EV1). These well-resolved maps refined in 3D to a 2.4–2.9 Å global resolution, allowing us to confidently assign and build nearly all $C_2$ and accessory domains, apart from the flexible N-terminal $C_2A$ domain (Figs. 1B–D and EV1; Appendix Figs. S2–S5 and Tables S1–S4). Myoferlin interacts with the MSP2N2 nanodisc through multiple binding motifs, all projecting on one side and forming several defined protein-lipid contact sites (Figs. 1D, EV1 and EV2). As expected, the remaining lipid membrane is largely disordered in these structures (Figs. 1D and EV2).

## Complex tertiary interfaces organize the lipid-bound state of myoferlin

Although predicted to adopt an extended, "beads on a string"-like topology (Dominguez et al, 2022; Leclère and Dulon, 2023), the overall cryo-EM maps of the lipid-bound myoferlin revealed a

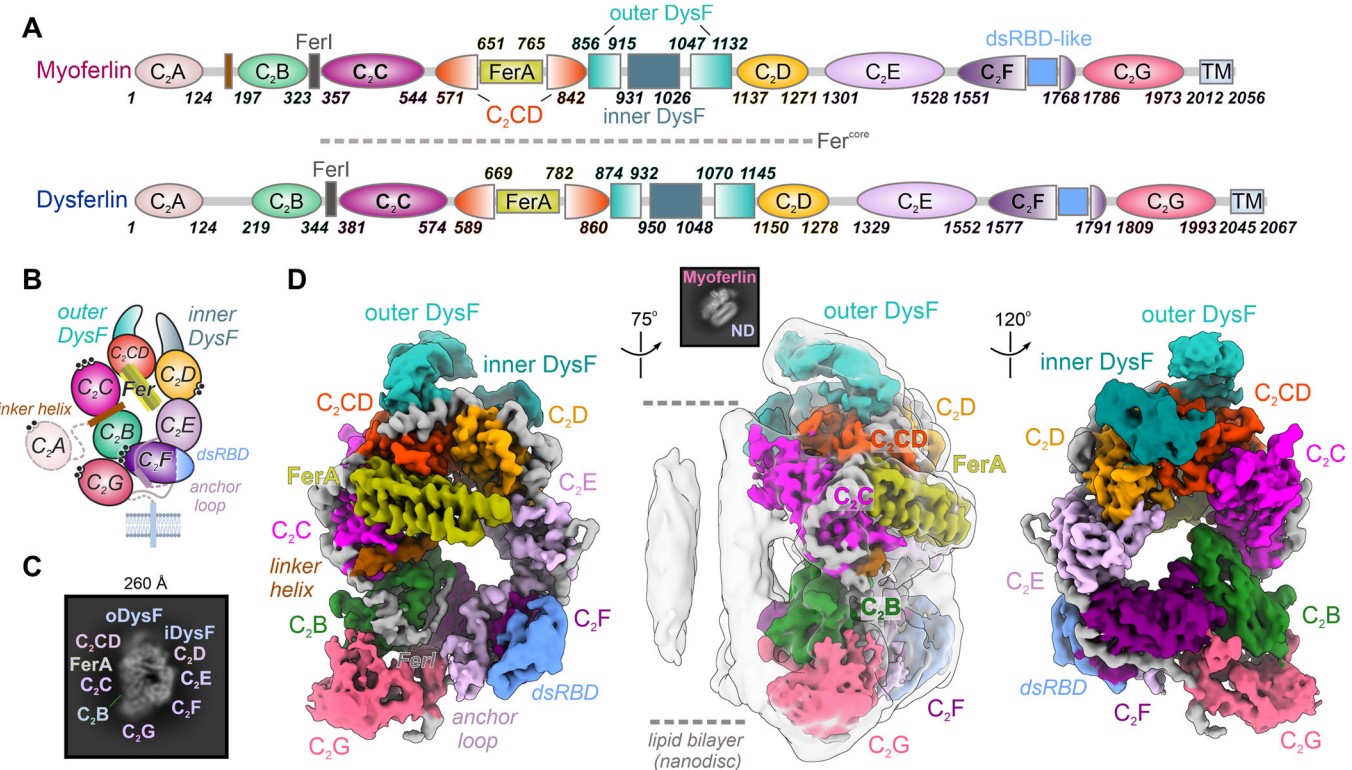

**Figure 1.  Cryo-EM structure of human myoferlin bound to a model lipid bilayer (nanodisc).**

(A) Schematic depicting the structure-based domain composition of human myoferlin and dysferlin. The $C_2$ domains of myoferlin and dysferlin are shown as coloured ellipses, whereas the DysF, FerA, and the transmembrane region (TM) are displayed as boxes. The linker helix between $C_2A$ and $C_2B$ is depicted as a brown box, whereas the remaining linker regions are coloured grey. dsRBD-like: the double-stranded RNA-binding-like subdomain of $C_2F$. (B) Schematic representation of the soluble myoferlin-lipid nanodisc cryo-EM structure. The transmembrane helix is depicted for orientation purposes and is absent from the soluble expression construct. The modelled $Ca^{2+}$ ions are indicated as black dots. (C) Typical top-view of lipid-bound soluble myoferlin. Myoferlin domains, which are visible in the 2D class average, are indicated. (D) The cryo-EM map of lipid-bound soluble myoferlin displayed in three different orientations. The individual ferlin domains are colour-coded as in A. The soluble myoferlin map has been locally scaled and low-pass filtered to 2.8 Å, whereas the nanodisc density (middle panel, contoured in grey around the ferlin cytosolic region) has been low-passed to 8 Å resolution to allow visualisation of the ordered lipid regions (see also Appendix Figs. S2–S5 and Fig. EV2). The inset (middle panel) shows a typical side-view class average, consistent with an asymmetric membrane recognition mechanism.

surprisingly compact domain architecture (Fig. 1B,D). As observed in 2D class averages (Fig. 1C; Appendix Figs. S2B, S4C and S5B,E), the individual structural motifs of myoferlin are distributed in an almost coplanar manner around a ~30 Å central cavity, describing an elliptic ring that spans ~150 Å and ~90 Å along its long and short axes, respectively (Fig. 2A). One side of the composite ring engages the membrane bilayer at four distinct contact sites, covering the entire nanodisc perimeter (Figs. 1D, 2A, and EV2). In contrast, the solvent-facing side of myoferlin harbours no lipid recognition motifs, consistent with the presence of a single membrane-binding surface (Figs. 1D and EV2A–F).

The top region of the ferlin ring (Figs. 1A,B and 2A) is formed by the structurally more rigid $C_2C$ (residues 357–544), $C_2CD$-FerA (residues 571–842), and $C_2D$ (residues 1137–1271) domains, which we refer to as the ferlin core module (Fer$^{core}$, Fig. EV1B,E and Movie EV2). The inner and outer DysF domains (Figs. 1D and 2A), each comprising two long β-strands, are closely linked to the Fer$^{core}$, through $C_2D$ and $C_2CD$, respectively, and separated by ~14 Å (Fig. EV1F). Opposite of the Fer$^{core}$, the N-terminal $C_2B$ (residues 197–323), the C-terminal $C_2F$ (residues 1551–1768), and the membrane-proximal $C_2G$ (residues 1786–1973) approach each

other in 3D and pack closely, despite being separated by more than 1200 residues (Figs. 2A,B and EV1, and Movie EV2). Unlike the Fer$^{core}$ motifs, these $C_2$ domains are engaged in fewer and transient interdomain contacts, in part mediated by loop regions or unstructured elements (Fig. 2B,C and Movie EV2). $C_2B$ contributes to the largest number of contacts and is surrounded from three sides by the top loops of $C_2F$ (L1, L3, and L4), the long β6-β7 subdomain of $C_2G$ (residues 1906–1945, Fig. EV1J,K) and the $C_2C$ domain of the Fer$^{core}$ (Fig. 2B,C). Additional contacts are provided by the membrane-binding β-hairpin of $C_2G$ (residues 1859–1879), inserted between the β4-β5 strands, and the upstream linker helix (residues 173–190), stacked between $C_2B$ and $C_2C$ (Figs. 2B,C, EV1A, and EV1J–L). The conserved but largely unstructured FerI motif (residues 324–356) appears to fold between $C_2F$, $C_2G$, and $C_2B$, before reaching $C_2C$, likely also stabilizing the composite interface (Figs. 2A,B and EV1A). Finally, the top and bottom arches of the ferlin ring are connected through the $C_2E$ domain, which interacts with both $C_2F$ and $C_2G$ through its extended anchor loop (residues 1447–1498), as well as with the upstream $C_2D$ through its loop 4 (L4, Figs. 2A, 2D,E, and EV1H).

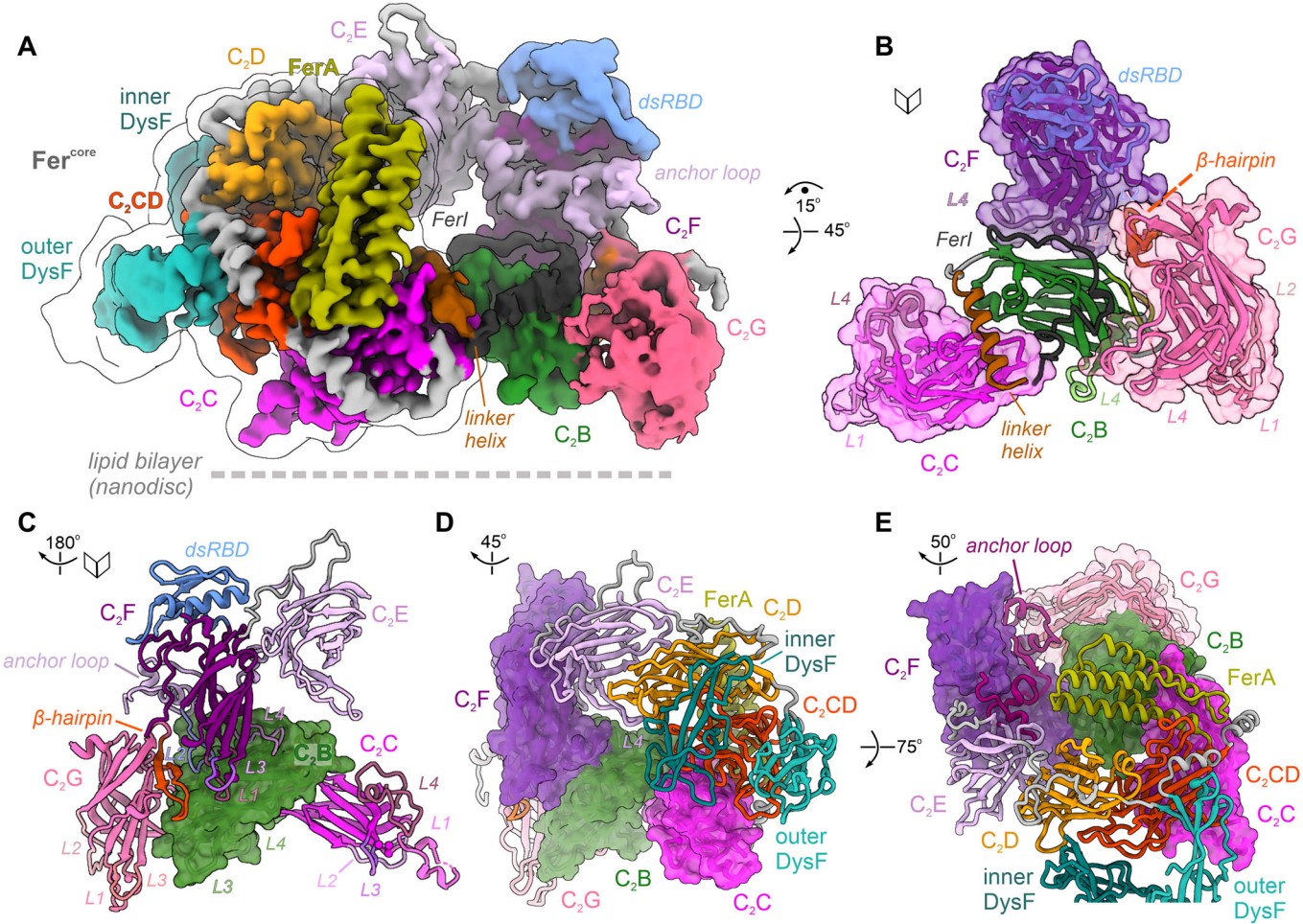

**Figure 2. The multipartite structural organization of lipid-bound myoferlin.**

(A) Overall cryo-EM map of lipid-bound soluble myoferlin (residues 1–1997), viewed from its N-terminal side. The map has been colour-coded after the modelled subunits and the key structural motifs are indicated. The Fer$^{core}$ module of myoferlin (the $C_2C$-$C_2D$ region) is contoured in light grey. (B) The multiple tertiary interfaces between the N-terminal $C_2B$-$C_2C$ and the C-terminal $C_2F$-$C_2G$, observed in the lipid-bound myoferlin structure. Additional structural elements, such as the linker helix and the FerI motif, bridging the $C_2B$ and $C_2C$ domains, and the β-hairpin subdomain of $C_2G$, are indicated. The $C_2$ domains, except $C_2B$, are depicted as transparent solvent-excluded surfaces, with the domain model fitted inside. (C) The $C_2B$ domain orientation in the lipid-bound state appears to be induced by its shared interfaces with $C_2C$, $C_2F$, and $C_2G$. $C_2B$ is shown in a surface representation. The top loops (L1-L4) of the interacting domains are indicated. (D) The peripheral inner and outer DysF motifs of myoferlin, viewed from the membrane-interacting side. The $C_2C$, $C_2F$, and $C_2G$ domains are depicted as surfaces. Note that the two DysF motifs are connected to the Fer$^{core}$ module through $C_2D$ (orange) and $C_2CD$ (red). (E) $C_2E$ connects the Fer$^{core}$ module to the C-terminal $C_2F$ domain through a long insertion loop (the "anchor loop"). $C_2E$ is depicted as a cartoon, whereas the $C_2B$-$C_2C$ and $C_2F$-$C_2G$ are shown as surfaces. The anchor loop inserts between the β6 and β7 strands of $C_2E$ (see also Fig. EV1H). At the same time (Fig. 2D), the L4 loop of $C_2E$ is oriented towards $C_2D$.

## The rigid ferlin core module is stabilized by a new $C_2$-like accessory domain

Consistent with limited proteolysis experiments in cultured cells (Woolger et al, 2017), the multipartite organization of the myoferlin ring appears to be centred around the tightly packed Fer$^{core}$ module (Fig. 3A,B and Movie EV2). Fer$^{core}$ covers almost one half of the ferlin ring and is distributed symmetrically at the two ends of the arch (Fig. 3A,B). In between $C_2C$ and $C_2D$, four distinctive motifs are observed (Fig. 3A,B): (i) the FerA domain (residues 651–765), located in the proximity of $C_2C$ and folded as a four-helix bundle (Harsini et al, 2018); (ii) the outer DysF domain (also known as N-DysF, residues 856–915 and 1047–1132), occupying a more central location; (iii) the inner DysF

domain (or C-DysF, residues 931–1027), inserted between the two β-strands of the outer DysF (Patel et al, 2008; Sula et al, 2014) and oriented towards the membrane surface; and (iv) a previously undetected $C_2$-like motif, which we denoted as the $C_2CD$ domain (residues 571–650 and 766–842).

The newly identified $C_2CD$ domain is present in all ferlins (Dominguez et al, 2022), including dysferlin and otoferlin (Figs. 3C and 6C,D; Appendix Figs. S6–S8, S9A–D, and S10A–D), and spans the distance between $C_2C$ and $C_2D$, appearing to "glue" the Fer$^{core}$ module together. Like all other $C_2$ domains of myoferlin, $C_2CD$ consists of two β-sheets and has a type-II topology (Dominguez et al, 2022). However, unlike typical $C_2$ domains, its second β-sheet comprises only three β-strands (Fig. 3C; Appendix Fig. S10B), the top loops (L1-L4) are extended, lack conservation, and engage in

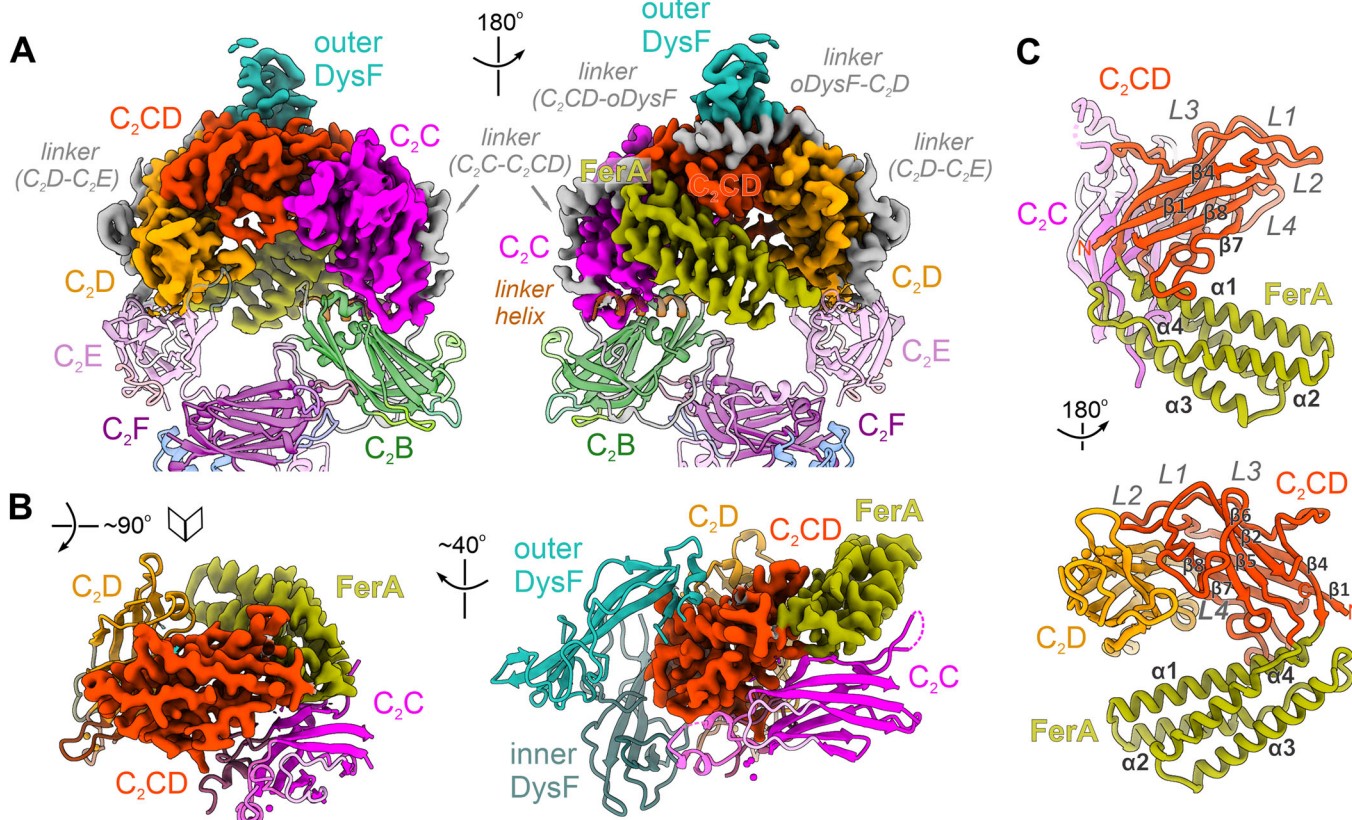

**Figure 3. Structure of the rigid Fer^core module.**

(A) Structure of the Fer^core module, spanning the top ferlin arch. Fer^core comprises the $C_2C$-$C_2D$ region (colour-coded cryo-EM map after the modelled subunits) and includes a new $C_2$-like structural domain, the $C_2CD$ domain (red). The neighbouring domains, $C_2B$ and $C_2E$-$C_2F$, are shown in cartoon representation. The modelled linker regions, connecting the $C_2$ domains, are coloured grey. (B) Interaction interfaces between $C_2CD$ and its neighbouring FerA, $C_2C$, and $C_2D$ domains. $C_2CD$ and FerA are depicted as cryo-EM maps, whereas $C_2C$, $C_2D$, outer and inner DysF are shown as cartoons. (C) Structure of myoferlin's $C_2CD$ and FerA domains. The seven β-strands of $C_2CD$ and the four α-helices of FerA are indicated.

tertiary contacts with $C_2C$, the inner DysF, and $C_2D$, instead of being available for $Ca^{2+}$ and/or phospholipid binding (see also Appendix Fig. S9E–G). $C_2CD$ is closely associated with the four-helix bundle of FerA, inserted between its β4-β5 strands (Fig. 3C; Appendix Fig. S10A,B). Interestingly, FerA does not engage the lipid nanodisc but instead exits the $C_2CD$ domain opposite from the membrane-binding surface (Fig. 3B,C, see also Appendix Figs. S9E–G and S11), partially closing the central cavity of myoferlin as it aligns diagonally along the $C_2C$-$C_2E$ axis without reaching $C_2E$. We suggest that the large contact interfaces of $C_2CD$ with the remaining Fer^core motifs (~4580 Å²) indicate a primary role as a repurposed $C_2$ domain packing platform (Appendix Fig. S11A–C), contrasting the $Ca^{2+}$-sensing and lipid-binding functions of other ferlin $C_2$ domains (see also Figs. 4A and 5).

## The composite membrane-binding interface of human myoferlin

Despite numerous structural studies of other $C_2$-domain proteins, such as synaptotagmins (Corbalan-Garcia and Gomez-Fernandez, 2014; Rizo, 2022; Rizo et al, 2022; Schauder et al, 2014; Seven et al, 2013), there has not been direct experimental evidence for how

multi-$C_2$ domain proteins interact with lipid bilayers through the combined binding of $Ca^{2+}$ and phospholipids. In particular, the relative orientation of all the membrane-binding domains, their lipid-interacting motifs, and their sensitivity to $Ca^{2+}$ (or lack thereof), remained speculative (Arac et al, 2006; Honigmann et al, 2013; Rizo et al, 2022; Seven et al, 2013). The cryo-EM structure of human myoferlin elucidates the organization of its seven $C_2$ and accessory domains upon concerted membrane-binding, enabling us to identify the precise motifs involved in lipid recognition.

In the nanodisc-bound structure (Fig. 4A,D,G), the cytosolic region of myoferlin engages the lipid surface primarily through the $C_2B$-$C_2G$, $C_2C$, the inner DysF, and $C_2F$-$C_2G$ domains, establishing a total of four membrane anchoring points. At the first binding site, we observe that the N-terminal $C_2B$ (Fig. 4B) faces the lipid membrane with its concave surface at a ~28° tilt angle. In this side-orientation, $C_2B$ projects the top loop 3 (L3) nearly perpendicular to the membrane plane, binding the bilayer through several hydrophobic (L271, A273) and polar (S270, R272) residues located at the L3 tip. The remaining residues of the L1-L3 loops do not participate in $Ca^{2+}$-coordination, as no ion densities were observed in the cryo-EM maps (Fig. EV1A). Instead, they establish contacts with $C_2G$, which contributes to the same contact site, suggestive of

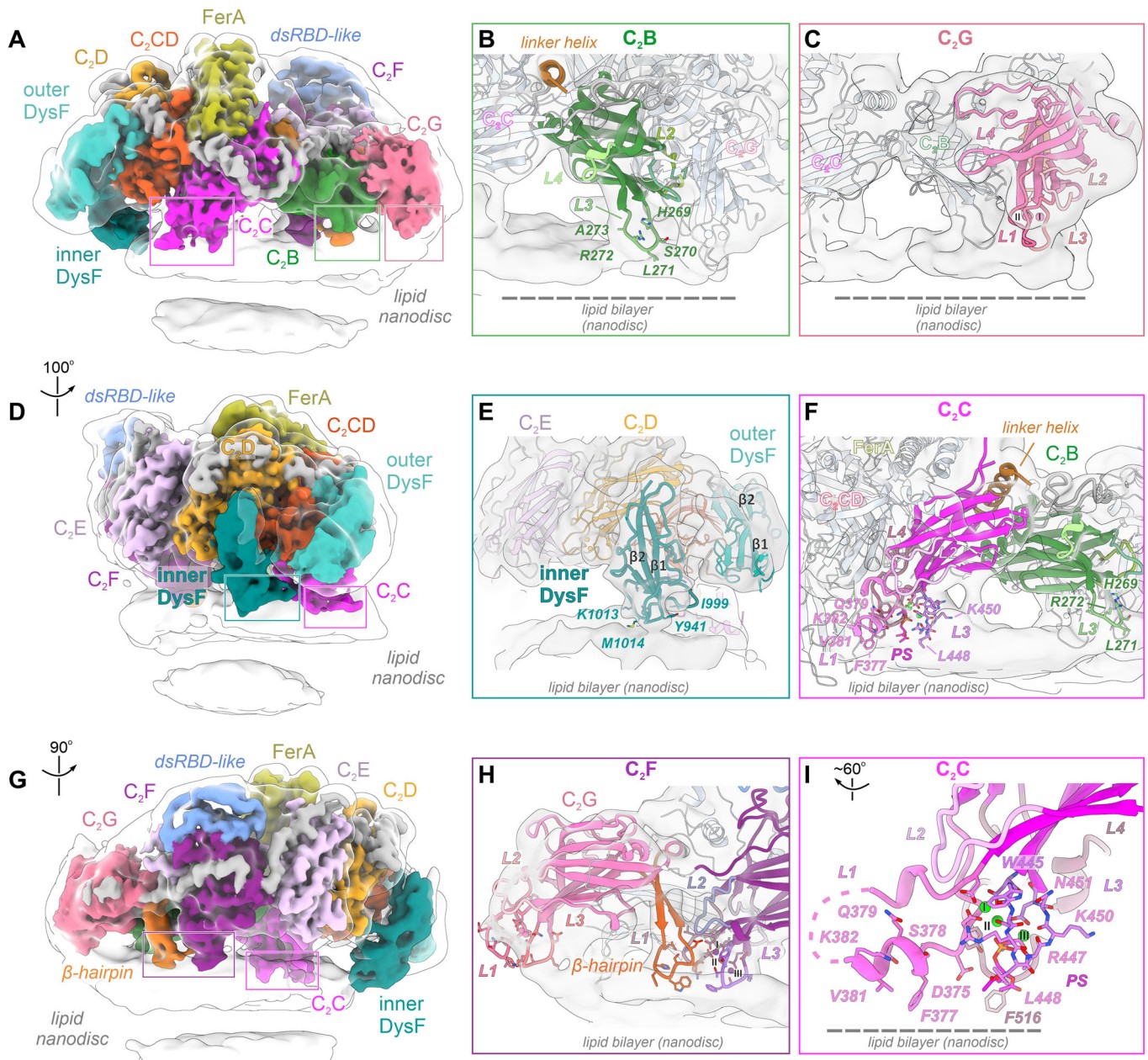

**Figure 4. The composite lipid membrane recognition interface of human myoferlin.**

(A) The lipid nanodisc contact interfaces of the $C_2B$ and $C_2C$ domains. The focused myoferlin (1–1997) cryo-EM map (colour-coded after the modelled subunits) is shown together with the 8 Å low-passed overall map (Appendix Figs. S2C and S5C). The ordered lipid nanodisc regions interact with myoferlin at multiple contact sites, defined by $C_2$ ($C_2B$, $C_2C$, $C_2F$, $C_2G$) and accessory (inner DysF, β-hairpin of $C_2G$) domains. (B) $C_2B$ interacts with lipid bilayers independent of $Ca^{2+}$ binding. Although no bound $Ca^{2+}$ ions were observed and $C_2B$ lacks conservation of the typical aspartate residues involved in divalent cation coordination, the domain engages the lipid bilayer (nanodisc) through its L3 top loop. $C_2B$ residues directly interacting or located close to the lipid bilayer are shown as sticks. (C) Close-up of $C_2G$'s lipid nanodisc contact interface. The top loops of $C_2G$ (L1 and L3) project close to $C_2B$'s binding site. The two modelled $Ca^{2+}$ ions are indicated. (D) The peripheral inner DysF motif engages the lipid bilayer close to $C_2C$'s binding site. The lipid nanodisc density is coloured in grey, whereas the cryo-EM map of myoferlin is coloured-coded as in (A). (E) Close-up of the inner DysF domain interacting with the lipid nanodisc. Inner DysF's loop residues located close to the nanodisc surface are shown as sticks, for orientation. Note that the outer DysF and $Ca^{2+}$-bound $C_2D$, flanking the inner DysF, do not bind the lipid nanodisc. (F) $C_2C$ establishes extensive, both $Ca^{2+}$-dependent and independent, contacts with the lipid nanodisc. $C_2C$ binds the lipid membrane through hydrophobic and basic residues of the L1, L3, and L4 loops, as well as through $Ca^{2+}$-mediated phospholipid headgroup coordination. The modelled phosphatidylserine (PS) and key interface residues are depicted as sticks. (G, H) The C-terminal $C_2F$ and $C_2G$ interact with the lipid nanodisc through a composite interface. The key lipid-binding motifs (the L3 loops of $C_2F$ and $C_2G$, the β-hairpin of $C_2G$) are indicated and the membrane-facing residues are shown as sticks. (I) The top L1 loop of $C_2C$ interacts with the lipid bilayer through an amphipathic insertion helix. Note the virtually parallel orientation of L1's amphipathic helix, engaging the nanodisc surface with its hydrophobic side (see also Figs. 4F and 5A–C). The $Ca^{2+}$-coordinating residues and the modelled PS are shown as sticks. The three $Ca^{2+}$ ions are marked with Roman numerals (I–III).

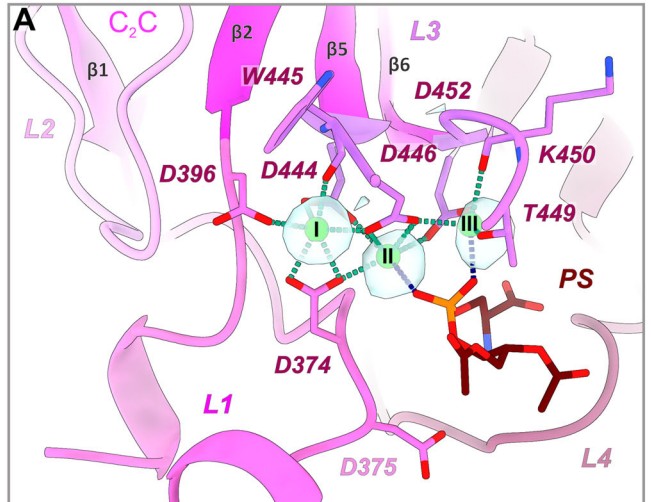
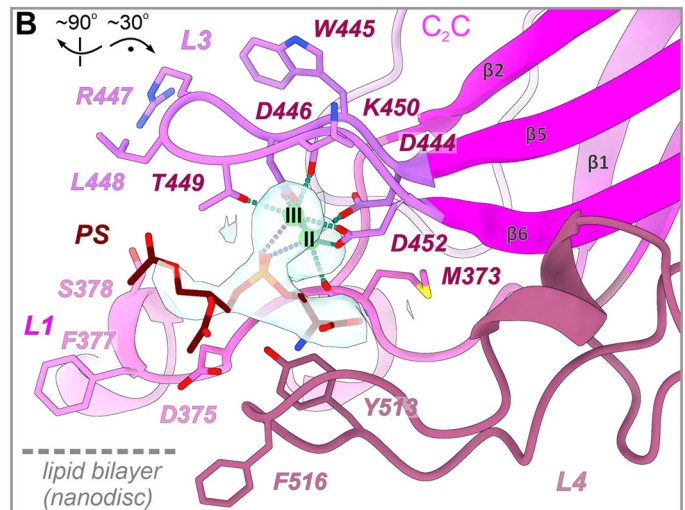
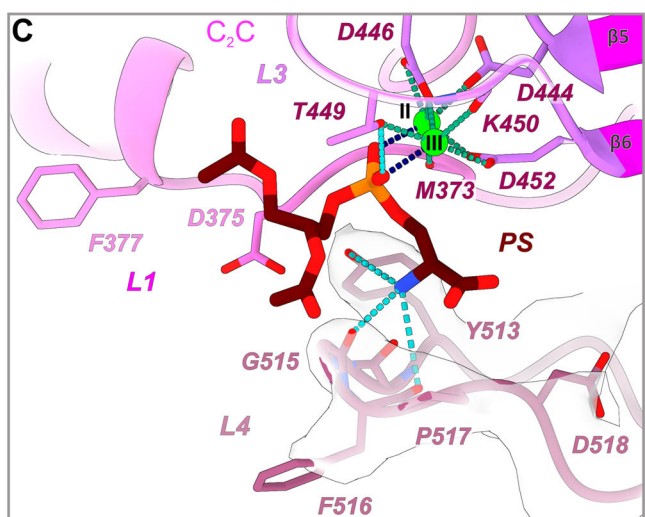
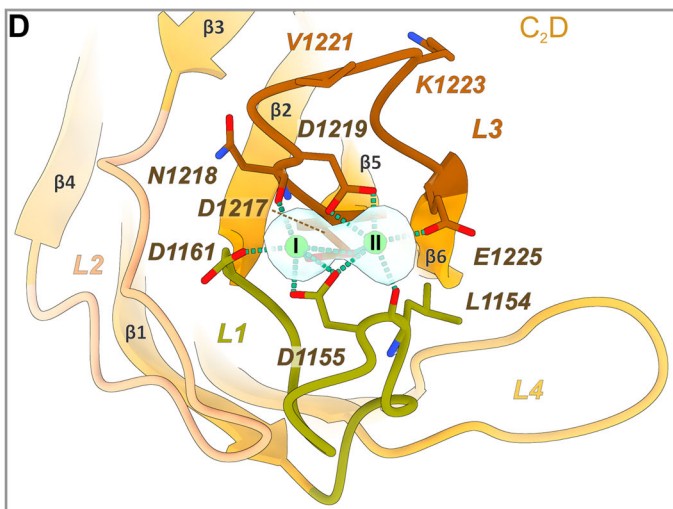
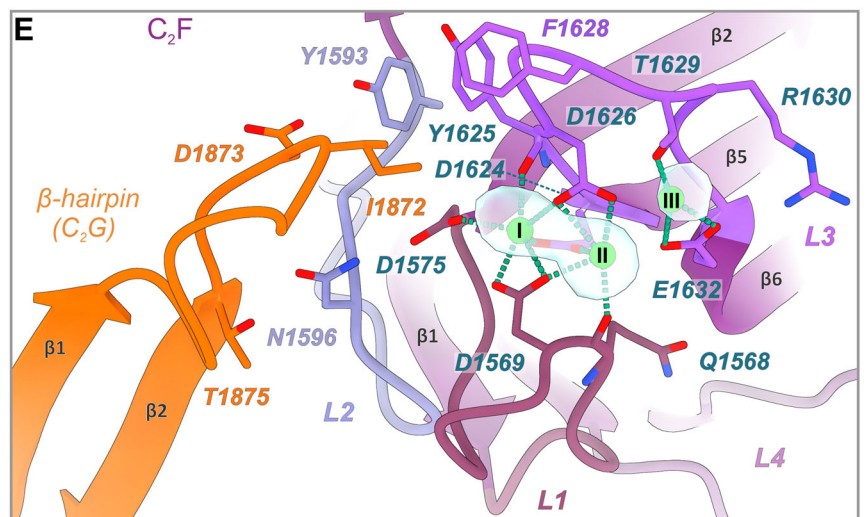
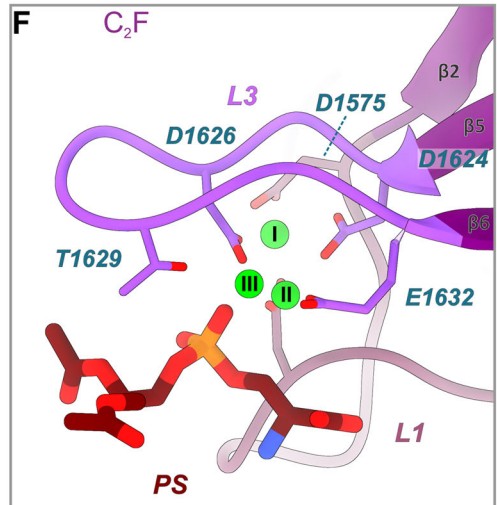

**Figure 5. Ca²⁺- and phospholipid-binding sites modelled in the lipid-bound structure of myoferlin.**

(A) Three Ca²⁺ ions and a phosphatidylserine (PS) molecule bind within the acidic pocket of C₂C. The normalized cryo-EM difference map (cyan, 10σ) is contoured around the modelled Ca²⁺ ions. The bound Ca²⁺ ions are labelled with Roman numerals (I–III). Ca²⁺ I and Ca²⁺ II have hexadentate coordination. Ca²⁺ III is coordinated by both L3 loop residues and a coordinating oxygen of PS. Coordination bonds are depicted as dashed lines. (B) The modelled PS headgroup contributes to Ca²⁺ ion coordination in C₂C. The cryo-EM difference map (cyan, 3.5σ) is contoured around PS and two Ca²⁺ ions (Ca²⁺ II and Ca²⁺ III). See also Fig. EV1M. (C) Molecular recognition of the PS headgroup by myoferlin's C₂C domain. The modelled PS is shown as sticks (dark red). Ca²⁺ coordination bonds are depicted as dark blue dashed lines, while polar interactions between PS and the L3-L4 loops of C₂C are coloured cyan. The cryo-EM density contoured around the L4 loop is displayed in grey. (D) Ca²⁺-binding sites of the C₂D domain. The cryo-EM difference map (cyan, 6.5σ) is contoured around the modelled Ca²⁺ ions. The two Ca²⁺ ions in C₂D (Ca²⁺ I and Ca²⁺ II) have hexadentate coordination achieved through interactions with residues from both the L1 and L3 loops. (E) Ca²⁺-binding sites of the C₂F domain. The cryo-EM difference map (cyan, normalized, 5σ) is contoured around the three modelled Ca²⁺ ions (Ca²⁺ I–III). C₂F residues contributing to Ca²⁺ coordination are shown as sticks and coloured teal. Several hydrophobic (Y1625, F1628) and basic (R1630) residues, located at the tip of the L3 loop, engage the lipid nanodisc (see also Fig. 4H). (F) A PS molecule is recruited to C₂F's Ca²⁺-binding loops. Myoferlin residues and the modelled Ca²⁺ ions (Ca²⁺ I–III) are represented as in (E). The Ca²⁺ III ion is coordinated by both the L3 loop and an oxygen atom of the PS phosphoryl group.

cooperative membrane binding (Figs. 4B,C and 2B,C). Like C₂B, the C-terminal C₂G faces the nanodisc with its concave side tilted by ~70° (Fig. 4C,H), directly interacting with a small membrane patch through both hydrophobic and basic residues of L1 and L3 (I1807, K1892, F1893, L1895). However, in contrast to C₂B, the acidic residues in L1 and L3 are conserved and likely involved in the coordination of two Ca²⁺ ions (Figs. 4C and EV1J–L).

Surprisingly, our structures revealed that an additional membrane anchoring point (Fig. 4D,E) is provided by the inner DysF motif. Located between C₂CD and C₂D, the inner DysF forms direct membrane contacts (Figs. 4D,E and EV1F), likely through hydrophobic (I997, I999, P1000, P1001, M1014), basic (K1013, H1016), and aromatic residues (Y941, Y1015), providing the first evidence, to our knowledge, of its role as a generic lipid-binding motif (Kaur et al, 2023). The positioning of the inner DysF at the membrane surface appears to be influenced by interactions with the L1-L2 loops of C₂CD and the L3 loop of the neighbouring C₂D (Figs. 2D,E and 4D,E). While engaging the inner DysF, C₂D coordinates two Ca²⁺ ions via its L1 and L3 loops (Figs. 4E, 5D and EV1G), despite being ~20 Å from the nanodisc surface. In addition, the Ca²⁺-bound L1 loop of C₂D interacts with the long L4 of the downstream C₂E (Fig. 2D,E), suggesting that C₂D's interfaces with both the inner DysF and C₂E may be sensitive to Ca²⁺ binding. Therefore, C₂D may simultaneously guide the inner DysF to the membrane while promoting the recruitment of C₂F-C₂G via C₂E, possibly in a Ca²⁺-dependent manner (see also Fig. 7D,E).

In conclusion, and consistent with our analyses (Appendix Fig. S1D–G), the established membrane contacts of C₂B, C₂G, and the inner DysF are neither exclusively electrostatic nor hydrophobic. Moreover, they do not appear to confer strict specificity for acidic phospholipids (such as PS and PI(4,5)P₂) through Ca²⁺-binding. Consequently, the positioning of these domains at the membrane may be indirectly modulated by neighbouring C₂ domains, such as C₂D, rather than by bona fide Ca²⁺-driven membrane recruitment.

## Myoferlin's Ca²⁺- and phospholipid-binding sites

Compared to the side-oriented C₂B and C₂G, the C₂C domain establishes a major membrane contact site by engaging the lipid nanodisc through its Ca²⁺-coordinating top loops (Figs. 4F,I and 5A–C). The local resolution of the myoferlin map enabled accurate modelling of three bound Ca²⁺ ions and a recruited PS molecule, revealing an intricate membrane recognition mechanism (Figs. 4F and 5A–C). The extended L1 loop forms the largest

contact interface between C₂C and the nanodisc. While coordinating two Ca²⁺ ions through D374 and D396 at its base (Figs. 5A–C and EV1B,C), the distal end of L1 comprises a conserved helical insertion (residues 377–382, Fig. 4I) of amphipathic nature, which interacts with the membrane via its hydrophobic side (F377, V381). The L3 loop of C₂C (Fig. 5A–C) coordinates all three Ca²⁺ ions through D444, W445 (backbone carbonyl), D446, T449, D452, and simultaneously binds the nanodisc (Figs. 4F and 5A–C) via several hydrophobic and polar residues at its tip (R447, L448, K450). Intriguingly, the modelled PS headgroup (Figs. 5B,C and EV1B,C), originating from the lipid nanodisc, binds at the interface between the L3 and the L4 loops of C₂C in all membrane-bound myoferlin structures (Figs. EV1M and EV2). As observed in other C₂ domains (Corbalan-Garcia and Gomez-Fernandez, 2014; Hirano et al, 2019; Honigmann et al, 2013; Rizo and Sudhof, 1998; Verdaguer et al, 1999), the recruited PS completes the coordination shells of the second and third Ca²⁺ sites. At the same time, its seryl moiety (Fig. 5B,C) is recognized by L4 residues (Y513, T514, G515), while L4 itself forms several additional membrane contacts (via F516 and P517).

Like C₂C, the C₂F domain engages the nanodisc surface through its Ca²⁺-coordinating loops, while also recruiting the β-hairpin subdomain of C₂G to the same site (Figs. 4G,H and 5E,F). In this configuration, the tip of the L3 loop inserts into the lipid bilayer (residues F1628, T1629, and R1630), contributing, along with the L1 loop, to the coordination of three Ca²⁺ ions (Fig. 5E,F). The first Ca²⁺ ion is coordinated by aspartate residues in L1 (D1575, D1569) and L3 (D1624), while coordination of the second and third Ca²⁺ ions involves a glutamate residue in L3 (E1632), a backbone carbonyl (R1630), and a recruited PS molecule (Fig. 5E,F). Interestingly, although not interacting with Ca²⁺ or phospholipids, the L2 of C₂F forms close contacts with the β-hairpin motif of C₂G, binding the same membrane patch (Fig. 4H). The β-hairpin subdomain (residues 1859–1879), inserted between the β4-β5 strands of C₂G, is conserved among ferlins (Dominguez et al, 2022) and comprises two antiparallel β-strands connected by a loop. Consistent with it representing a lipid-binding accessory motif (Figs. 4H and EV1J), the β-hairpin loop interacts tightly with the nanodisc through hydrophobic (F1869, W1870, I1872) and basic residues (K1866, H1868), possibly alongside the L3 loop of C₂F and the adjacent core of C₂G.

Altogether, membrane recognition by human myoferlin is achieved through both Ca²⁺-dependent and Ca²⁺-independent mechanisms, involving four C₂ domains (C₂B, C₂C, C₂F, and

$C_2G$) and two accessory elements (the inner DysF and the β-hairpin of $C_2G$). The lipid-interacting motifs are asymmetrically arranged on one side of the ferlin ring and form multiple composite interfaces ($C_2B$-$C_2G$, $C_2F$-β-hairpin), highlighting a unique binding mode for this large multi-$C_2$ domain protein.

## Lipid-free structures of human myoferlin and dysferlin

To understand how ferlins are organized prior to membrane recruitment and to clarify which conformational rearrangements accompany their membrane binding, we obtained additional cryo-EM structures of $Ca^{2+}$-bound soluble myoferlin (residues 1–1997) and the closely related dysferlin (residues 1–2017), at overall resolutions of ~3.2 Å and ~3.5 Å, respectively (Fig. 6A–F, Appendix Figs. S6–S8, and Fig. EV3). In the lipid-free state, both ferlins adopt a generally similar ring-like structure, formed by only six $C_2$ domains (Fig. 6A–F and Movie EV3). The rigid $Fer^{core}$ module is well resolved (Appendix Figs. S9A–C, S10–S11) and has a similar organization in both ferlin structures, including the $C_2C$, $C_2CD$-FerA, and $C_2D$ domains (Appendix Fig. S8A–G; Fig. S9D–G). Importantly, in both structures (Fig. EV3F; Appendix Fig. S12A–C), $C_2C$ and $C_2D$ coordinate two divalent cations each, and alanine substitutions of seven aspartates in their top loops abolished the $Ca^{2+}$- and membrane-binding activity of the $Fer^{core}$ (Appendix Figs. S12D–G and S13A–C). The inner and outer DysF motifs are linked to the $Fer^{core}$ and are folded as in other myoferlin structures (Figs. 6C,F and EV3C–E; Appendix Fig. S8K). In contrast, the membrane-interacting $C_2B$ and $C_2F$-$C_2G$ are dynamic and adopt different conformations in the lipid-free state (Figs. 6E,F and EV3A; Appendix Figs. S6A and S7G–J).

Compared to the nanodisc-bound structures (Figs. 1D and EV2A–F), in the lipid-free cryo-EM models of both dysferlin (Fig. 6C,D; Appendix Figs. S7–S9) and myoferlin (Figs. 6E,F and EV3A–E), the N-terminal $C_2B$ and the C-terminal $C_2E$ domains adopt a virtually antiparallel orientation, in which the top loops of $C_2B$ project towards the solvent-facing side of the ferlin ring (Fig. 6F). The $C_2B$ and $C_2E$ domains also establish several small tertiary interfaces with $C_2C$ and $C_2D$ of the $Fer^{core}$ module, respectively, as well as with the $C_2F$ domain (both $C_2B$ and $C_2E$), aligned along the short axis of the ring (Figs. 6C,D,F; Appendix Fig. S7G–J). At the same time, like in the lipid-bound state, the anchor loop of $C_2E$ wraps around $C_2F$, ~40 Å away from $C_2E$'s β-sheets (Fig. 6C,F). As a result of these multiple contacts, the $C_2F$ domain appears surrounded on three sides by $C_2B$, $C_2E$, and the poorly resolved and dynamic $C_2G$ domain (Figs. 6D–F and EV3D,E; Appendix Fig. S7G–J). This conformation of $C_2F$ differs significantly from the membrane-bound state of myoferlin, where the domain engages the lipid bilayer through its $Ca^{2+}$-binding loops (Figs. 4G and 6F). Consequently, in the lipid-free state (Figs. 6C–F and EV3D,E), while still available for $Ca^{2+}$-binding, the orientation of $C_2F$'s L1-L3 loops seems incompatible with efficient membrane insertion, possibly due to steric hindrance from the interacting $C_2B$ domain.

To confirm the physiological relevance of the lipid-free ferlin state, resolved here at lower-resolution for myoferlin and dysferlin, we subjected soluble dysferlin (residues 1–2017) to chemical cross-linking in the presence of $Ca^{2+}$ and identified the cross-linked residues by tandem mass-spectrometry (Cretu et al, 2016). Mapping the cross-linked lysines onto the cryo-EM structure of

dysferlin (Fig. 6G,H; Appendix Fig. S14A–C, and Dataset EV1) indicated that 66 out of 71 observed unique cross-links (~92.96%) occurred between residues less than 35 Å apart (48/52, ~92.31%, when omitting the more dynamic $C_2G$ domain). Importantly, the existence of characteristic interfaces between $C_2B$, $C_2E$, $C_2F$-dsRBD, and $C_2G$, as observed in the lipid-free dysferlin model (Fig. 6H), was strongly supported by the cross-linking data, with only five outlier interdomain cross-links being observed, all of them involving the flexible $C_2G$ and the anchor loop of $C_2E$. Consistent with the dysferlin cryo-EM map (Fig. 6G,H; Appendix Fig. S14B; Fig. S14D), our cross-linking data placed the $C_2E$ domain between $C_2B$, $C_2D$, and the $C_2F$-dsRBD module. Furthermore, the location of $C_2F$-dsRBD at the base of the dysferlin ring was supported by its cross-links to both $C_2B$ and $C_2E$, as well as to the neighbouring $C_2G$, flanking $C_2F$ from both sides (Fig. 6H; Appendix Fig. S14B). From these analyses, we conclude that the lipid-free conformation is indeed populated in solution, possibly representing the default state of ferlins in cells.

## The different conformational states of human ferlins

Despite being comparable in general terms, the lipid-free and membrane-bound ferlin structures represent two distinct conformations (Fig. 7A and Movie EV4). Structure-based superposition and modelling of the transition between the two resolved states of myoferlin allowed us to decipher the possible sequence of events (Fig. 7B–E, Movies EV4 and EV5). We delineated major structural rearrangements occurring at several N-terminal ($C_2B$) and C-terminal ($C_2E$-$C_2G$) sites of the myoferlin ring, indicating a multistep transition (Fig. 7B–E and Movie EV5). Compared to the rigid $Fer^{core}$, our analyses show that the N-terminal $C_2B$ appears to rotate by ~160° upon membrane binding and moves by ~11 Å towards the membrane plane (Fig. 7B–E and Movie EV5). This dramatic change in the domain's orientation entails both disruption ($C_2E$), reconfiguration ($C_2C$ and $C_2F$), and formation (with $C_2G$ and the linker helix) of new tertiary interfaces and membrane contacts (through $C_2B$'s L3). Concomitantly, our superposition indicates that the $C_2F$-dsRBD module, which, in the lipid-free state, interacts with $C_2B$, appears to translocate by ~20 Å and further rotate by an additional ~38° from the ring's periphery towards the $Fer^{core}$ (Fig. 7C). The large-scale displacement of the $C_2F$ domain would enable the previously hindered $Ca^{2+}$-binding loops to directly engage the lipid bilayer (Fig. 7C). Accompanying this $C_2F$ repositioning (Fig. 7D and Movie EV5), the upstream $C_2E$, interacting with $C_2F$ through the anchor loop, also rotates by ~32° in the direction of the $Fer^{core}$, while its long L4 contacts the $Ca^{2+}$-bound $C_2D$. Notably, the new poses of $C_2E$ and $C_2F$ appear to be stabilized initially by the repositioned $C_2B$ (Fig. 7C,D) and the unstructured FerI motif, which covers the distance between $C_2B$ and $C_2C$. Finally, the $C_2G$ domain, which is flexible in the lipid-free states, binds the reconfigured $C_2B$-$C_2F$ interface on both sides to close the myoferlin ring. Besides establishing new interdomains contacts, the rearranged $C_2G$ simultaneously engages the lipid membrane with its $Ca^{2+}$-binding loops and β-hairpin subdomain, inserting into the bilayer in the vicinity of $C_2B$ and $C_2F$, respectively (Fig. 7E). Interestingly, our structures show that $C_2G$ binds more tightly phosphoinositide-containing bilayers, possibly because of additional interactions between its polybasic patch and exposed $PI(4,5)P_2$ headgroups, as previously suggested and/or

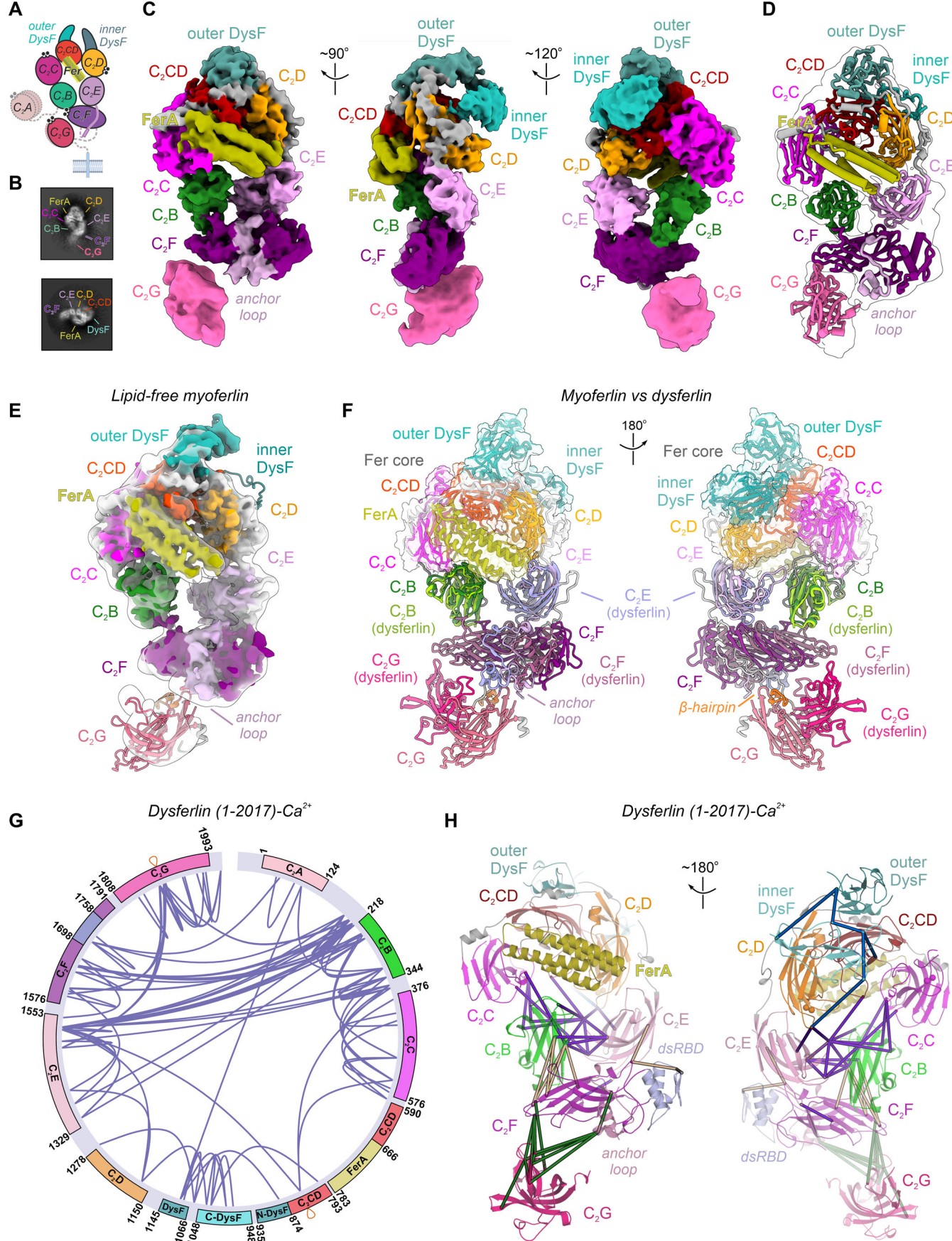

◄ **Figure 6. Cryo-EM structures of human dysferlin and myoferlin in their lipid-free states.**

(A) Schematic representation of dysferlin's cryo-EM structure. The transmembrane region, absent from the expressed cytosolic domain of dysferlin (residues 1–2017), is shown for orientation. (B) Two representative, reference-free 2D class averages of dysferlin. The visible dysferlin domains are indicated. (C) The overall cryo-EM map of lipid-free, $Ca^{2+}$-bound dysferlin (1–2017), rendered in three different orientations. The individual ferlin domains are coloured as in A. The map has been locally scaled and low-pass filtered to 5 Å (see also Appendix Fig. S6–S8). (D) The cryo-EM model of dysferlin (1–2017) showing the observed tertiary interfaces between $C_2B$, $C_2E$, $C_2F$, and $C_2G$. The final model is fitted within the overall map. (E) The cryo-EM structure of lipid-free, $Ca^{2+}$-bound myoferlin (1–1997). The consensus map of myoferlin in the lipid-free state has been scaled and coloured after the modelled domains. The map has been fitted into an overall map (lowpass filtered to 10 Å, see also Figs. EV3A,E) to visualize the dynamic $C_2G$ domain. (F) Structural superposition of myoferlin (1–1997) and dysferlin (1–2017) in their lipid-free states. The structures were aligned based on the rigid Fer$^{core}$ module (the $C_2C$-$C_2D$ region, depicted as a solvent-excluded surface), which is similarly organized. The $C_2B$ and $C_2E$-$C_2G$ domains are shown in different colours. Compared to the C-terminal $C_2F$-$C_2G$, the $C_2B$ and $C_2E$ domains adopt largely similar poses in the two lipid-free structures (see also Fig. 6C, D). (G) Circular plot showing all observed intramolecular cross-links between dysferlin's structural motifs. The BS3 chemical cross-linking data, at FDR (false discovery rate) of 1%, were not filtered by score or spectral count (see also Appendix Fig. S14A–C and Dataset EV1). (H) Selected interdomain BS3 cross-links mapped onto the lipid-free structure of dysferlin. The displayed Euclidean distances (coloured lines) were calculated between the Cα atoms of the cross-linked lysine residues and are all below the 35 Å theoretical distance threshold (see also Appendix Fig. S14A,B and Dataset EV1).

observed in other $C_2$ domain structures (Carpenter et al, 2023; Guerrero-Valero et al, 2009; Guillen et al, 2013; Kwok et al, 2023; Padmanarayana et al, 2014) (Fig. EV4A–C).

The conformational rearrangement of myoferlin to its lipid-bound state is possible because the $C_2$ domains undergoing displacement share relatively small contact interfaces in the lipid-free conformation, and the connecting linker regions are long and partially unstructured (Figs. 1A and EV2B; Appendix Fig. S13D). For example, $C_2E$'s contact interfaces with $C_2B$ (~252 Å²), $C_2D$ (~347 Å²) and $C_2F$ (~235 Å², excluding the anchor loop) are equally small. Likewise, in the lipid-free state, the $C_2B$-$C_2F$ interface involves a relatively small number of residues (~193 Å²) and is not present in all the imaged particles (Figs. 6D,E and EV3A; Appendix Fig. S6C). Moreover, calpain-cleavage of myoferlin and alternative dysferlin isoforms appears to release of the C-terminal $C_2F$-$C_2G$ domains, which indicates that the $C_2B$-$C_2F$ and $C_2E$-$C_2F$ interfaces are also dynamic in cells (Piper et al, 2017; Redpath et al, 2014). In this respect, the accessory ferlin structural motifs—the β-hairpin of $C_2G$, the linker helix, and FerI—resolved exclusively in the lipid-bound state of myoferlin, are likely required to stabilize the reconfigured $C_2$ domain interfaces.

Therefore, the tertiary interactions between the Fer$^{core}$, $C_2B$, $C_2E$ and $C_2F$, observed in the lipid-free structures, seem to be generally weak, having possibly evolved to facilitate efficient membrane sampling on fast timescales and in a complex cellular environment. Formation of new and extended contacts between $C_2B$, $C_2F$, and the repositioned $C_2G$ not only increases the structure's stability, but also allows their concomitant, in-plane binding as part of a composite and asymmetric interface. Because the C-terminal $C_2G$ is connected to the transmembrane helix through a relatively short linker (approximately 20 residues), our structures indicate that the large-scale movement of the domain towards the Fer$^{core}$ would shorten the distance between the ferlin vesicle and the target membrane, likely facilitating their close apposition. Indeed, supporting such a scenario, we observe that both myoferlin and dysferlin engage lipid bilayers in a $Ca^{2+}$-dependent manner and promote tight vesicle-vesicle interaction (docking) in vitro, when both PS and PI(4,5)P₂ are present on the target membrane (Fig. EV4D–G). Given the high degree of conservation of their lipid-free structures, it is likely that dysferlin and, possibly, otoferlin transition through a similar sequence of domain rearrangements and conformational states upon membrane binding (Fig. 8). Such a role of ferlins in progressing from loose vesicle tethering to tight

docking of membranes has been suggested for otoferlin based on electron tomography (Vogl et al, 2015) and, like myoferlin (Fig. EV4D,E) and dysferlin (Codding et al, 2016), otoferlin's $C_2$ domains accelerate SNARE-dependent membrane fusion in vitro (Johnson and Chapman, 2010). These functional similarities indicate that ferlins may indeed share common organization principles and a conserved membrane remodelling mechanism.

## Discussion

Ferlins are the largest and the most complex multi-$C_2$ domain $Ca^{2+}$-sensitive factors (Lek et al, 2012). In this study, we report the near-atomic cryo-EM structures of human myoferlin in its $Ca^{2+}$ and lipid-bound state. We also obtained cryo-EM models of both myoferlin and the related dysferlin in their lipid-free states and carried out supporting experiments to validate our structural models. These analyses provide first insights into how the unique structural features of ferlins are adapted to their many ascribed roles in cellular trafficking pathways, underlying the acute response to membrane and organelles injuries, muscle biogenesis and metabolism, or fast transmitter exocytosis at hair cell ribbon synapses.

We propose that the multifaceted roles of ferlins in diverse $Ca^{2+}$-dependent pathways in cells could stem from their ability to sample alternative conformational states (Fig. 8), achieved by exploiting a unique structural organization. Contrary to previous models (Dominguez et al, 2022; Woolger et al, 2017; Xu et al, 2011), the ferlin $C_2$ domains are not trivially organized as "beads on a string" or as in other, well-studied multi-$C_2$ domain factors (Schauder et al, 2014; Shin et al, 2005). Instead, the conserved structural motifs pack uniquely in 3D and form state-defining interfaces, bridging both neighbouring and sequence-distant domains. The central ferlin core module (Fer$^{core}$), comprising the region between the $C_2C$ and $C_2D$ domains, likely, maintains part of the, otherwise dynamic, ferlin structure rigid and might support the ordered and reversible $Ca^{2+}$-/lipid-driven conformational transitions of ferlins. In this respect, the newly identified ferlin $C_2$-like domain, the $C_2CD$ domain, observed in our myoferlin and dysferlin structures, and the closely linked four-helix bundle of FerA might have evolved to promote the tight packing of the Fer$^{core}$ module. Importantly, AlphaFold predictions support the presence of a similarly organized Fer$^{core}$ in all remaining paralogs, including otoferlin,

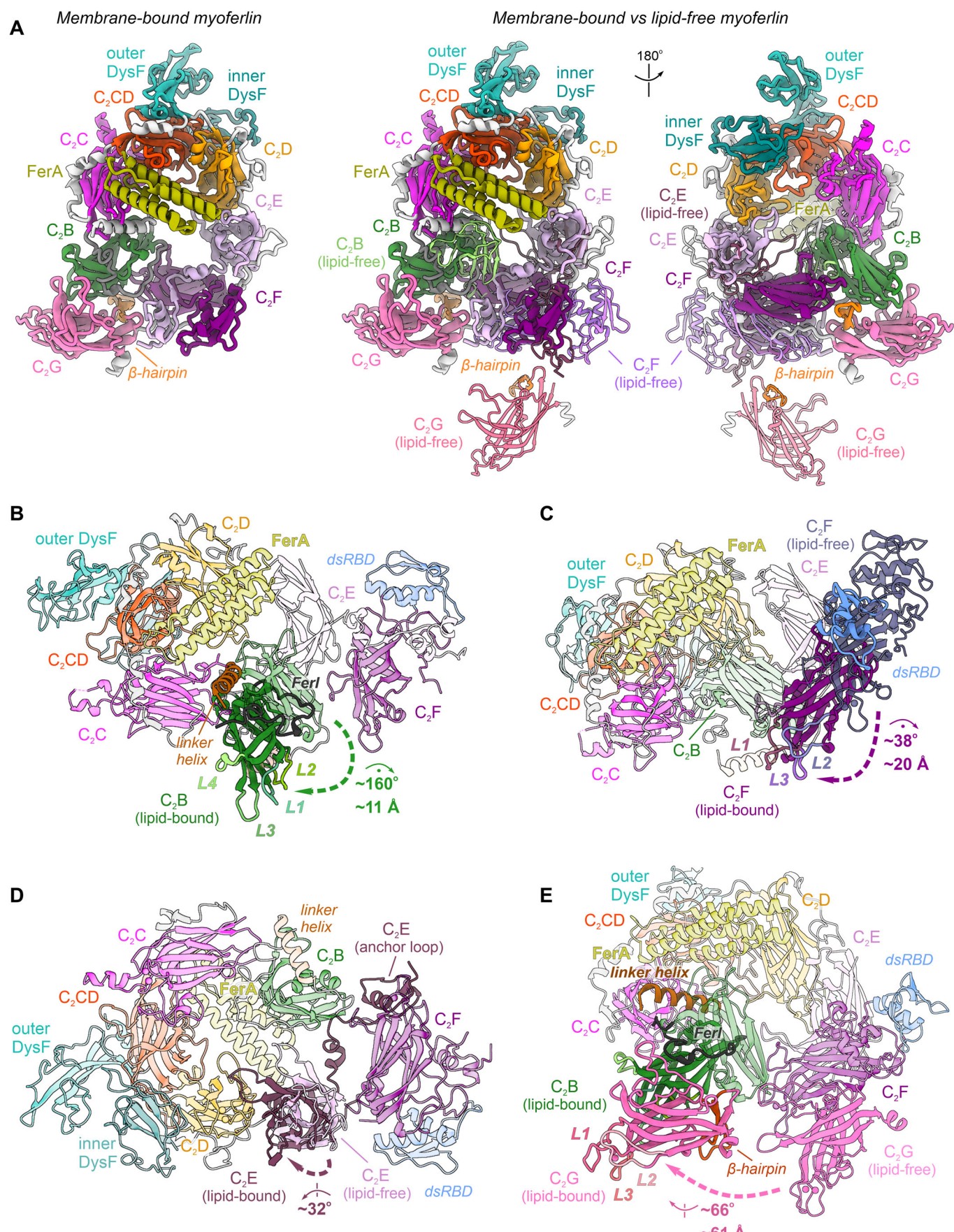

**Figure 7.  Large-scale conformational transition of ferlins upon lipid membrane binding.**

(A) Side-by-side comparison of lipid-free and membrane-bound myoferlin. The structures were superimposed based on the Fer$^{core}$ module, and the rearranged domains are shown in different colours. (B) The large-scale conformational rearrangement of C$_2$B upon lipid membrane binding. Myoferlin in the lipid-free state is shown as a transparent model. As a result of a ~160° out-of-plane rotation and an ~11 Å translation, the top L3 loop of C$_2$B projects towards the nanodisc surface in the lipid-bound state. Concomitantly, the linker helix and the FerI motif are positioned between C$_2$C and C$_2$F-C$_2$G, likely stabilizing the new pose of C$_2$B. (C) Large-scale displacement of the C$_2$F domain upon lipid nanodisc binding. Compared to the lipid-free states of myoferlin and dysferlin, where the top loops are oriented towards C$_2$B, C$_2$F moves by ~20 Å in the direction of Fer$^{core}$ and the membrane plane. (D) In-plane rotation of C$_2$E during lipid membrane recognition by human myoferlin. Because of this rearrangement, the L4 loop of C$_2$E establishes contacts with the Ca$^{2+}$-bound L1 loop of C$_2$D, likely fixing the new pose. As the anchor loop of C$_2$E wraps around C$_2$F, the movements of C$_2$E and C$_2$F may be coupled. (E) C$_2$G translocates by ~61 Å to engage the lipid nanodisc, together with C$_2$B and C$_2$F, in the myoferlin structure. The β-hairpin subdomain of C$_2$G is coloured orange. Since C$_2$G is flexible and does not interact with C$_2$B and C$_2$F in the lipid-free ferlin structures, it is likely that the movements of C$_2$B and C$_2$F precede the recruitment of the membrane-proximal C$_2$G domain. The myoferlin model in the lipid-free state is shown as a transparent cartoon.

whose mutations cause the nonsyndromic deafness DFNB9 and are a target of the first inner ear gene-therapy trials (Lv et al, 2024; Moser et al, 2024) (Appendix Fig. S15A,D).

Although appearing defined and stable enough to be observed upon cryo-EM imaging, the interfaces between the membrane-proximal domains and the Fer$^{core}$ module are more dynamic in the lipid-free states of myoferlin and dysferlin. This notable structural feature appears to be a direct consequence of the highly specialized roles served by ferlins in cells. Having low energy barriers between their discrete conformations could, possibly, explain the transiently formed contacts between state-defining C$_2$ domains, such as C$_2$B, C$_2$F and C$_2$G (Fig. 8). This may constitute a significant advantage in protein-rich microenvironments, such as the active zones of hair cell ribbon synapses (otoferlin) (Moser et al, 2020) or caveolae-rich plasma membrane compartments (dysferlin) (Corrotte et al, 2013), allowing them to sample conformations close to the ground state. As a result, switching between the different ferlin conformations could be accomplished by reversibly shifting the equilibrium towards a given, functionally relevant state without a significant free energy consumption (Fig. 8). Consistently, all ferlin-dependent cellular processes occur on fast timescales, depend on Ca$^{2+}$, and are highly dynamic and reversible (Michalski et al, 2017; Pangrsic et al, 2010), with ferlins appearing to be needed for both the forward and reverse reactions.

An important question raised by our structural analyses pertains to the possible connections between the observed structural changes upon membrane binding in vitro (Fig. 8) and the documented roles of ferlins in promoting Ca$^{2+}$-sensitive vesicle tethering and membrane fusion (Codding et al, 2016; Johnson and Chapman, 2010; Marty et al, 2013; Vogl et al, 2015). As specialized trafficking factors highly expressed in muscle cells, the membrane-anchored myoferlin and dysferlin reside at the sarcolemma and shuttle to the endosomal compartment, without accumulating in a defined intracellular vesicle pool (Bansal et al, 2003; Davis et al, 2000; Doherty et al, 2005; Hofhuis et al, 2017; Hofhuis et al, 2020; Paulke et al, 2024). The related otoferlin is enriched at presynaptic active zone membranes and in the synaptic vesicles of inner hair cells, while also shuttling through the endosomal pathway during a synaptic vesicle release cycle (Chen et al, 2024; Jung et al, 2015; Moser et al, 2020; Pangrsic et al, 2012; Revelo et al, 2014). In all cases, the ability of ferlins to react to an increase in intracellular Ca$^{2+}$, following large sarcolemma injuries (dysferlin) or a sound-evoked receptor potential (otoferlin), appears to be determined by their cellular localization on vesicles, interactions with other factors, and the local phospholipid composition of the target

membranes. Consistent with having such multimodal mechanisms of action, the cryo-EM models of myoferlin and dysferlin revealed a highly dynamic structural organization, apparently evolved to accommodate different membrane environments and binding modalities (Fig. 8). Our structures show that the Ca$^{2+}$-sensitive N-terminal C$_2$A (in type-I ferlins) and the membrane proximal domains (C$_2$F-C$_2$G) are more dynamic in the lipid-free ferlin states, which could result in an increased membrane capture radius (Fig. 8). We, therefore, propose that by employing their mobile C$_2$ domains as pioneer Ca$^{2+}$ and lipid-sensitive motifs, ferlins would be able to efficiently sample and approach distant target membranes (Fig. 8). Consequently, membrane attachment of these pioneer C$_2$ domains would trigger, in the next step, the tight association of the remaining Fer$^{core}$ core domains (C$_2$C-C$_2$D and the inner DysF), altogether assembling an asymmetric ring-like structure that could bridge two cellular membranes (tethering) and, at the same time, promote local membrane remodelling through deep insertion into the bilayer at multiple contact sites (tight vesicle docking, Fig. 8). In our model, as only one side of the ring would engage the lipid bilayer, the available ferlin domains, such as FerA, could simultaneously interact with other pathway factors, such as SNAREs (Codding et al, 2016) or other ferlin molecules (Pangrsic et al, 2012), further narrowing the gap between the two membranes to facilitate their fusion (Fig. 8). This structural adaptation could be particularly relevant for dysferlin, whose functions in plasma membrane repair have been linked to the Ca$^{2+}$-sensitive clustering of highly heterogenous endomembrane vesicles to promote their fusion (Bansal et al, 2003; Codding et al, 2016; Cooper and McNeil, 2015; Han et al, 2012).

Future functional and structural studies are needed to probe the validity of the proposed mechanisms, beyond simple nanodisc membranes, particularly within the context of pathway-defining, higher-order ferlin complexes (Grushin et al, 2019; Stepien et al, 2022; Zhou et al, 2015; Zhou et al, 2017). These studies will likely entail the use of alternative imaging modalities (cryo-electron tomography and super-resolution imaging) and computational approaches (molecular dynamics simulations), applied to both surrogate membrane systems (liposomes) and in vivo triggered vesicle fusion (Chakrabarti et al, 2022; Held et al, 2024; Imig et al, 2020; Kovtun et al, 2020; Rizo et al, 2022; Wang et al, 2024). Such complementary analyses will help exclude improbable non-physiological myoferlin conformations induced by nanodiscs and further elucidate the roles of additional ferlin molecules and ferlin-SNAREs interactions in the vesicle-membrane fusion cycle.

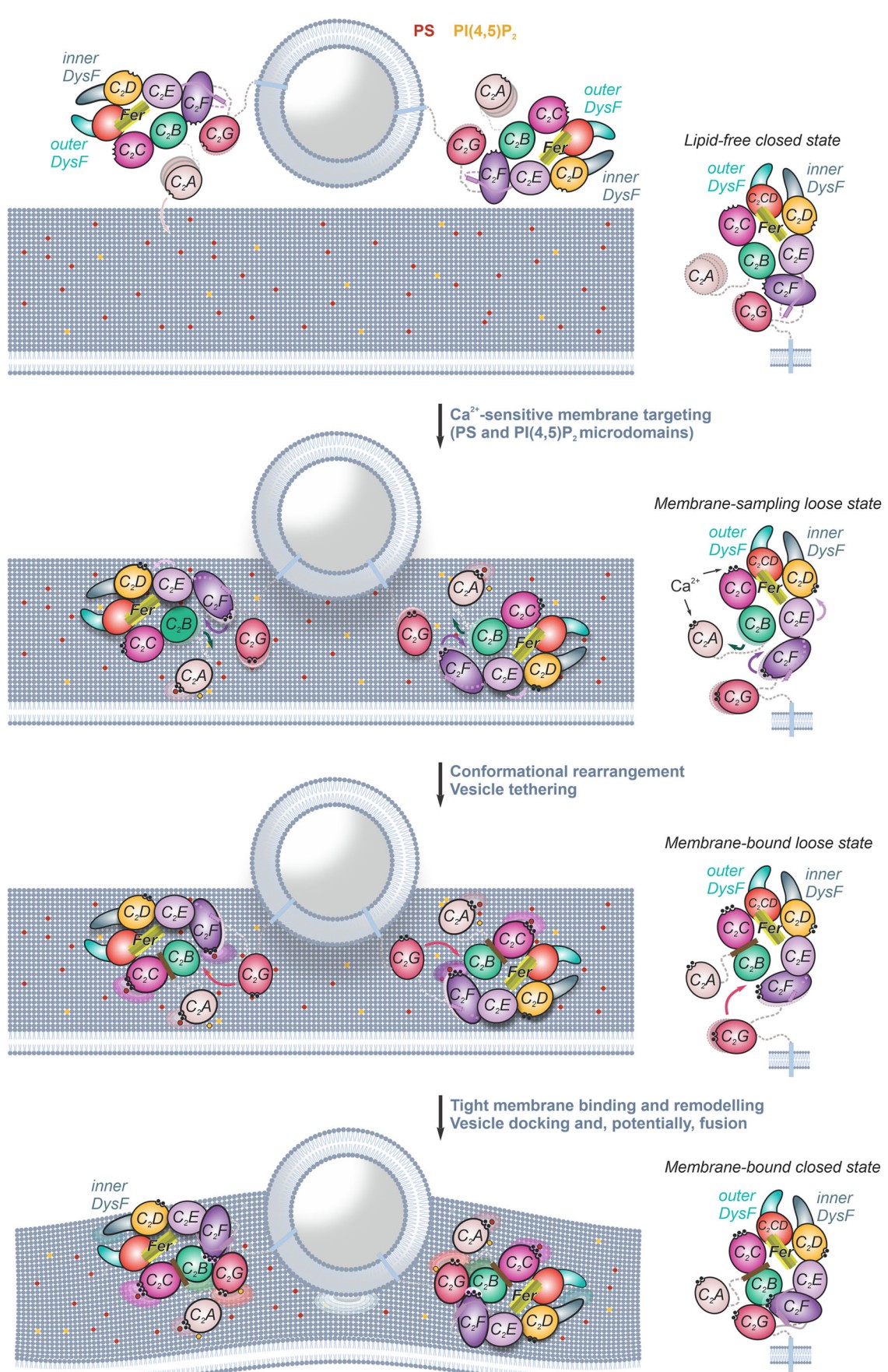

**Figure 8. Model of ferlins' Ca²⁺-sensitive recruitment and binding to lipid membranes.**

Schematic depicting a structure-guided model of how ferlins facilitate Ca²⁺-dependent vesicle targeting, docking, and local membrane remodelling to modulate and possibly promote fusion of two lipid bilayers. The ferlin domains are colour-coded, with those interacting with lipid membranes highlighted. PS (Phosphatidylserine) and PI(4,5)P₂ (Phosphatidylinositol-4,5-bisphosphate) are indicated in red and orange, respectively. The membrane-sampling loose state has been observed in the lipid-free dysferlin cryo-EM data (Appendix Figs. S6A and S14A). The hypothetical membrane-bound loose state, where the C-terminal C₂G domain is yet to engage the lipid bilayer, is based on AlphaFold2 predictions of ferlins (Appendix Fig. S15B–D). The lipid-free and membrane-bound closed states are depicted based on the cryo-EM structures of lipid-free myoferlin/dysferlin and membrane-bound myoferlin, respectively.

# Methods

### Reagents and tools table

| Reagent/Resource | Reference or Source | Identifier or Catalog Number |
|---|---|---|
| **Experimental models** | | |
| High Five (*Trichoplusia Ni*, BTI-Tn-5B1-4) insect cells | Thermo Fisher Scientific Expression Systems | cat#B85502 cat#94-002F |
| Sf9 (*Spodoptera frugiperda*) insect cells | Thermo Fisher Scientific | cat#11496015, cat#12659017 |
| NEB Stable Competent *E. coli* (High Efficiency) cells | New England Biolabs | cat#C3040H |
| DH10EMBacY (Multibac) electroporation-competent *E. coli* cells | Imre Berger lab (University of Bristol, Geneva Biotech) | Prepared in house |
| XL1-Blue electroporation-competent *E. coli* cells | Agilent | Prepared in house |
| **Recombinant DNA** | | |
| pCC10 (twin-StrepII-HRV3C-pFastBac) | This study, Schmitzova et al, 2023 | |
| pCC4 (twin-StrepII-HRV3C-Dysferlin (1-2080)-pFastBac): canonical isoform, UniProt O75923-1, codon optimized synthetic gene | This study | |
| pCC17 (twin-StrepII-HRV3C-Dysferlin (1-2017)-pFastBac): canonical isoform | This study | |
| pCC18 (10xHis-twin-StrepII-HRV3C-Dysferlin (1-2056)): human dysferlin alternative isoform 13 (UniProt O75923-13) | This study | |
| pCC26 (twin-StrepII-HRV3C-Dysferlin (214-2017)-pFastBac): canonical isoform | This study | |
| pCC48 (twin-StrepII-HRV3C-Dysferlin (1503-1560-GSGNGN (linker)-1613-2056)-pFastBac): minimal dysferlin, derived from the alternative isoform 13 of human dysferlin (residues 1-2119, UniProt O75923-13) | This study | |
| pCC75 (twin-StrepII-HRV3C-Dysferlin (214-1281)-pFastBac): canonical isoform, UniProt O75923-1 | This study | |

| Reagent/Resource | Reference or Source | Identifier or Catalog Number |
|---|---|---|
| pCC84 (twin-StrepII-HRV3C-Dysferlin (214-1281, D1168A/D1174A)-pFastBac): C₂D-DDA-1 | This study | |
| pCC85 (twin-StrepII-HRV3C-Dysferlin (214-1281, D1230A/D1232A)-pFastBac): C₂D-DDA-2 | This study | |
| pCC90 (twin-StrepII-HRV3C-Dysferlin (214-1281, D1168A/D1174A/D1230A/D1232A)-pFastBac): C₂D-QDA | This study | |
| pCC94 (twin-StrepII-HRV3C-Dysferlin (214-1281, D465A/D473A/D1168A/D1174A/D1230A/D1232A)-pFastBac): C₂C-DDA/C₂D-QDA | This study | |
| pCC95 (twin-StrepII-HRV3C-Dysferlin (214-1281, D417A/D465A/D473A/D1168A/D1174A/D1230A/D1232A)-pFastBac): C₂C-TDA/C₂D-QDA | This study | |
| pCDNA3.1-Myoferlin-HA | Addgene, Bernatchez et al, 2007 | cat#22443 |
| pCC68 (twin-StrepII-HRV3C-Myoferlin (1-2061)-pFastBac) | This study | |
| pCC76 (twin-StrepII-HRV3C-Myoferlin (1-1997)-pFastBac) | This study | |
| **Oligonucleotides and other sequence-based reagents** | | |
| PCR primers | This study, Microsynth Seqlab | Appendix Table S7 |
| **Chemicals, Enzymes and other reagents** | | |
| Phusion High-Fidelity DNA polymerase | New England Biolabs | Cat#M0530S |
| Restriction enzymes | New England Biolabs | SspI-HF (cat#R3132L) DpnI (cat#R0176L) |
| T4 DNA Polymerase (LIC-qualified) | Novagen | cat#70099 |
| T4 DNA Ligase | Thermo Fisher Scientific | cat#EL0011 |
| Purified human MSP2N2 (His-tagged) | Cube Biotech | cat# 26172 |
| High Pure Plasmid Isolation kit | Roche | cat#11754785001 |
| NucleoSpin Gel and PCR clean-up | Macherey-Nagel | cat#740609 |

| Reagent/Resource | Reference or Source | Identifier or Catalog Number |
|---|---|---|
| FuGENE HD | Promega | cat#E2311 |
| EDTA-free cOmplete protease inhibitors cocktail | Roche | cat#11836170001 |
| Glutaraldehyde | Electron Microscopy Sciences | cat#16220 |
| BS3 (Bis(sulfosuccinimidyl) suberate) | Thermo Fisher Scientific | cat#A39266 |
| DDM (n-Dodecyl-β-D-maltoside) | Glycon Biochemicals | cat#D97002 |
| OG (n-Octyl-β-D-glucopyranoside) | Glycon Biochemicals | cat#D97001 |
| LMNG (Lauryl maltose neopentyl glycol) | Anatrace | cat#NG310 |
| Lipids | Avanti Polar Lipids | DOPC, cat#850375P-200mg DOPE, cat#850725P-25mg DOPS, cat#840035P-25mg Cholesterol (ovine wool), cat#700000P Brain $PI(4,5)P_2$, cat#840046X-1mg, cat#840046P-1mg 18:1 Dansyl-PE, cat#810330C-5mg 18:1 Liss Rhod PE, cat#810150C-10mg 18:1 NBD PE, cat#810145C-5mg |
| Sf-900 III SFM | Gibco | cat#12658019 |
| ESF 921 protein free medium | Expression Systems | cat#96-001-01 |
| Strep-Tactin XT 4FLOW high-capacity resin | IBA Lifesciences | cat#2-5030-010 |
| HiTrap Q Sepharose HP column | Cytiva | cat#17-1154-01 |
| Superose 6 Increase 10/300 GL | Cytiva | cat#29-0915-96 |
| Superdex 200 Increase 10/300 GL | Cytiva | cat#28-9909-44 |
| Chelex 100 | Bio-Rad | cat#142-1253 |
| Bio-Beads SM-2 | Bio-Rad | cat#1523920 |
| One-Step Blue | Biotium | cat#21003 |
| NuPAGE 4–12% Bis-Tris polyacrylamide gels | Thermo Fisher Scientific | cat#NP0323BOX |
| Slide-A-Lyzer dialysis devices/cassettes | Thermo Fisher Scientific | MWCO 10 kDa, cat#69570 or cat#66383 MWCO 20 kDa, cat#69590 |
| NativeMark™ Unstained Protein Standard | Invitrogen | cat#LC0725 |
| Gel filtration calibration kit HMW | Cytiva (GE Healthcare) | cat#28-4038-42 |
| UltrAuFoil R1.2/1.3 300 gold mesh grids | Jena Bioscience | cat#X-201-AU300 |

| Reagent/Resource | Reference or Source | Identifier or Catalog Number |
|---|---|---|
| Quantifoil R1.2/1.3 200 copper mesh grid | Jena Bioscience | cat#X-101-CU200 |
| **Software** | | |
| EPU v.3.6 | Thermo Fisher Scientific | |
| cryoSPARC v4.1, v.4.4-v.4.5.1 | https://cryosparc.com | |
| RELION-5.0 | https://github.com/3dem/relion; Kimanius et al, 2024 | |
| ChimeraX v.1.6-v.1.8 | https://www.cgl.ucsf.edu/chimerax | |
| UCSF Chimera v.1.16 | https://www.cgl.ucsf.edu/chimera | |
| Phenix v1.19.2-4158, v.1.21.2-5419 | https://phenix-online.org; Afonine et al, 2018 | |
| CCP4 v.8.0 | https://www.ccp4.ac.uk | |
| CCPEM v.1.7.0 | https://www.ccpem.ac.uk | |
| Coot v.0.9.8.1 | Emsley and Cowtan, 2004 | |
| Pymol v.2.5.4 | Schrödinger LLC | |
| Isolde | Croll, 2018 | |
| LocScale v1, v2 | Jakobi et al, 2017 | |
| DeepEMhancer | Sanchez-Garcia et al, 2021 | |
| OriginPro 2020 v.9.7.0.188 | OriginLab | |
| GraphPad Prism v.9, v.10 | https://www.graphpad.com | |
| FluorEssence v.3.9 | Horiba Jobin Yvon | |
| PR.ThermControl v.2.3.1 | NanoTemper Technologies | |
| AcquireMP, DiscoverMP | Refeyn | |
| pLink v.2.3.9 | Chen et al, 2019 | |
| AlphaFold2 | Google DeepMind, Jumper et al, 2021 | |
| **Other** | | |
| Prometheus NT.48 | NanoTemper Technologies | |
| Fluorolog 3 spectrofluorometer | Horiba Jobin Yvon | |
| OneMP mass photometer | Refeyn | |
| Vitrobot Mark IV | Thermo Fisher Scientific | |
| Plasma Cleaner | Harrick Plasma | |
| Titan Krios G4 (Falcon 4i, SelectrisX) | Thermo Fisher Scientific | |

## Plasmids and molecular cloning

The codon-optimized full-length canonical human dysferlin (DYSF, FER1L1) expression construct (NM_003494.4, transcript variant 8, isoform 1, UniProt O75923-1) was constructed by GenScript and cloned into the pFastBac1 vector backbone using the EcoRI and KpnI restriction sites, in-frame with an N-terminal twin-StrepII affinity tag, which is cleavable with the HRV-3C (Human Rhinovirus 3C) protease. The full-length human myoferlin (MYOF, FER1L3, Uniprot Q9NZM1-1) was obtained from Addgene (pCDNA3.1-Myoferlin-HA, (Bernatchez et al, 2007)) and subcloned via ligation-independent cloning (LIC) into a modified pFastBac backbone (Schmitzova et al, 2023), in-frame with an N-terminal twin-StrepII tag, cleavable with HRV-3C. The soluble dysferlin (residues 1–2017) and myoferlin (residues 1–1997) constructs and the domain truncation mutants of dysferlin—Fer$^{core}$ (residues 214–1281) and minimal dysferlin (encompassing the $C_2F$-$C_2G$ domains, residues 1503–1560 and 1613–2056)—were obtained using "around-the-horn" PCR-based cloning with the full-length constructs as templates. Neutralizing substitutions in the $Ca^{2+}$-binding motifs of the dysferlin core region (residues 214–1281) were introduced through sequential PCR-based site-directed mutagenesis (Appendix Table S7). All constructs were verified by Sanger sequencing (Microsynth Seqlab GmbH, Göttingen) of the open reading frames (ORFs). Cloning primers were synthesized by Microsynth Seqlab GmbH (Göttingen).

## Expression and purification of human myoferlin and dysferlin

To enable their expression in insect cells, all myoferlin and dysferlin constructs were incorporated into bacmids through transformation of electro-competent DH10EMBacY (MultiBac) E. coli cells, as previously described (Cretu et al, 2018; Cretu et al, 2021; Cretu et al, 2016; Schmitzova et al, 2023). The bacmids were extracted for transfection using the High Pure Plasmid Isolation kit (Roche) and precipitated with isopropanol. To produce $V_0$ baculoviruses, adherent Sf9 (Spodoptera frugiperda) cells, cultured in Sf-900 III SFM (Gibco), were transfected with the prepared bacmids using FuGENE HD (Promega). The initial $V_o$ viruses were amplified by infecting Sf9 cells at a 1:10–1:20 ratio to produce the $V_1$ generation of baculoviruses. For large-scale protein production, $V_1$ baculoviruses were used to infect Sf9 or Hi5 (High Five, Trichoplusia Ni) suspension cultures, grown in the ESF 921 protein-free medium (Expression Systems), at titers sufficient to induce cell proliferation arrest after 24 h (Cretu et al, 2016; Schmitzova et al, 2023). Hi5 and Sf9 cells were typically infected at a density of $\sim 1.0 \times 10^6$ cells/mL and harvested 60–72 h post-infection, when cell viability dropped to ~80–85%. Insect cell infection was monitored every 16–24 h by observing the expression of the eYFP marker. Full-length and soluble dysferlin (1–2017) were overexpressed on a preparative scale in Hi5 cells. All myoferlin samples (full-length myoferlin (1–2061) and soluble myoferlin (1–1997)), the Fer$^{core}$ module of dysferlin and its mutant variants were expressed and purified from Sf9 cells.

All dysferlin and myoferlin constructs were purified by exploiting the highly specific twin-StrepII affinity tag and their binding to an anion-exchange resin (Appendix Fig. S1A). Membrane-anchored full-length myoferlin and dysferlin were purified in the presence of DDM (n-Dodecyl-β-D-maltoside, Glycon Biochemicals), whereas all soluble ferlin constructs were purified without detergent after cell lysis. Typically, insect cells from ~1 L culture were resuspended in 10–15 mL of lysis buffer per gram of cell pellet (50 mM HEPES-KOH, pH 7.5, 300 mM KCl, 10% (v/v) glycerol, 4 mM DTT (Dithiothreitol), 2–2.5% (w/v) DDM (or 0.2% (v/v) Triton X-100 (Roth) for soluble constructs) and cOmplete protease inhibitors (Roche), 1 tablet per 50 mL buffer). Membrane-anchored constructs were detergent-extracted by incubating the lysates at 4 °C for 90 min on a roller mixer. For soluble constructs, insect cells were lysed by sonication on ice using the Branson Ultrasonics Sonifier 250 (duty cycle: 30%, output: 3, sonication time: 2 min). The crude lysates were cleared by ultracentrifugation at 46,300 r.p.m. for 1 h at 4 °C in a Type 70 Ti rotor (Beckman Coulter) or at 15,000 r.p.m. for 1 h at 4 °C in a JA-18 rotor (Beckman Coulter). In the next step, the cleared lysates were filtered through a 0.8 μm Minisart membrane (Sartorius) and incubated for 1 h at 4–8 °C with 4–6 mL of Strep-Tactin XT 4FLOW high-capacity resin (50% slurry, IBA Lifesciences) per 1 L of culture. The resin was pelleted by centrifugation at 2000 r.p.m. and washed sequentially with the Wash buffer (50 mM HEPES-KOH, pH 7.5, 300–500 mM KCl, 5% (v/v) glycerol, 2 mM DTT, 0.5% (w/v) DDM (for membrane-anchored constructs)) and the Binding buffer (50 mM HEPES-KOH, pH 7.5, 150 mM KCl, 5% (v/v) glycerol, 2 mM DTT, 0.1% (w/v) DDM (for membrane-anchored constructs)). Bound proteins were eluted with the Elution buffer (50 mM HEPES-KOH, pH 7.5, 120–140 mM KCl, 5% (v/v) glycerol, 2 mM DTT, 0.1% (w/v) DDM (for the membrane-anchored constructs), 1 mM EDTA (Ethylenediaminetetraacetic acid), pH 7.5, 60 mM biotin). The affinity tag was not cleaved.

To remove nucleic acid contaminants and minor impurities, the Strep-Tactin eluates were purified by anion-exchange chromatography using a 5 mL HiTrap Q Sepharose HP column (Cytiva). The samples were applied to the column in the Buffer A (20 mM HEPES-KOH, pH 7.5, 150 mM KCl, 5% (v/v) glycerol, 2 mM DTT (or 1 mM TCEP (Tris(2-carboxyethyl)phosphine)), 0.03% (w/v) DDM (for membrane-anchored constructs)) and eluted using a 0–30% gradient formed between Buffer A and Buffer B (20 mM HEPES-KOH, pH 7.5, 1 M KCl, 5% (v/v) glycerol, 2 mM DTT (or 1 mM TCEP), 0.03% (w/v) DDM (for membrane-anchored constructs)) over 60–80 mL. The peak ferlin fractions were concentrated by ultrafiltration to ~3–3.5 mg/mL (membrane-anchored constructs), ~4.0–5.5 mg/mL (soluble dysferlin (1–2017) and myoferlin (1–1997)) or ~7.5–10 mg/mL (domain truncation mutants of dysferlin), snap frozen in liquid nitrogen, and stored at −80 °C. The identities of the purified proteins were verified by mass-spectrometry (Proteomics Facility, Max-Planck-Institute for Multidisciplinary Sciences, Göttingen).

## NanoDSF-based characterization of myoferlin and dysferlin

As a means of protein quality control, all myoferlin and dysferlin preparations (both soluble and membrane-anchored) were subjected to nanoDSF (nano differential scanning fluorimetry) measurements. In a typical nanoDSF assay, a 5 μL protein sample (~0.6 μM) was mixed with 5 μL assay buffer (25 mM HEPES-KOH, pH 7.5, 150 mM KCl and 0.03% (w/v) DDM for membrane-

anchored constructs), which had been pretreated with Chelex 100 (Bio-Rad) and supplemented with increasing concentrations of CaCl$_2$ or MgCl$_2$ (0–40 mM). After incubation for 10 min at room temperature, the samples were loaded into capillaries, and the emission intensity at 350 nm and 330 nm was measured as a function of temperature using a Prometheus NT.48 instrument (NanoTemper Technologies). The temperature was increased from 20 to 95 °C at an unfolding ramp of 1 °C/min. The excitation power was adjusted to yield at least 2000 integrated fluorescence counts at both wavelengths. The melting temperatures ($T_m$) were estimated by plotting the first derivative of the 350 nm/330 nm ratio as a function of temperature using PR.ThermControl v.2.3.1. The $[Me^{2+}]_{1/2}$ values were estimated in GraphPad Prism 10 (v10.3.1) by nonlinear regression fitting to a modified Hill function: $T_m([Me^{2+}]) = T_m i + (T_m f - T_m i) \times \frac{[Me^{2+}]^n}{([Me^{2+}]_{1/2}^n + [Me^{2+}]^n)}$, where $T_m i$ and $T_m f$ represent the initial and final $T_m$ of the titration series, respectively, and $n$ is the Hill coefficient.

## Preparation of liposomes

All lipids (phospholipids and cholesterol) used to prepare liposomes (LUVs, large unilamellar vesicles) and lipid nanodiscs were obtained from Avanti Polar Lipids, dissolved in chloroform to their working concentrations, and stored at −20 °C. Porcine brain PI(4,5)P$_2$ (L-α-phosphatidylinositol-4,5-biphosphate) was dissolved in a chloroform:methanol:water (20:9:1) solution. Unlabelled LUVs devoid of anionic phospholipids (referred to as "DOPC-DOPE-only LUVs") were prepared by mixing DOPC (1,2-dioleoyl-sn-glycero-3-phosphocholine, 18:1 (Δ9-cis) PC), DOPE (1,2-dioleoyl-sn-glycero-3-phosphoethanolamine, 18:1 (Δ9-cis) PE) and cholesterol (ovine wool) in a 7:2:1 molar ratio to a final concentration of 8 mM. LUVs containing DOPS (1,2-dioleoyl-sn-glycero-3-phospho-L-serine, referred to as "15 mol% DOPS LUVs") and both DOPS and PI(4,5)P$_2$ (referred to as "25 mol% DOPS/ 5 mol% PI(4,5)P$_2$ LUVs") were prepared by substituting a portion of DOPC in the lipid mixture with the respective anionic phospholipids to obtain the desired ratio. Dansyl-labelled LUVs were prepared by replacing a portion of DOPE in the lipid mixture with 5 mol% 18:1 Dansyl PE (1,2-dioleoyl-sn-glycero-3-phosphoethanolamine-N-(5-dimethylamino-1-naphtalenesulphonyl)).
Similarly, Rhodamine B- and NBD-labelled LUVs were obtained by replacing a portion of DOPE in the lipid mixture with 1 mol% 18:1 Lissamine Rhodamine B PE (1,2-dioleoyl-sn-glycero-3-phosphoethanolamine-N-(lissamine rhodamine B sulfonyl)) and 1 mol% 18:1 NBD PE (1,2-dioleoyl-sn-glycero-3-phosphoethanolamine-N-(7-nitro-2-1,3-benzoxadiazol-4-yl). To obtain unilamellar liposomes, lipids were transferred to a glass vial, the solvent was evaporated under a nitrogen stream, and the lipid film was dried in a vacuum desiccator (~200 mbar) for at least 3 h. The lipid film was hydrated in reconstitution buffer (20 mM HEPES-KOH, pH 7.5, 150 mM KCl) and extruded through a 0.4 μm Nuclepore track-etch membrane (Cytiva) at least 21 times using a Mini-extruder (Avanti Polar Lipids); these liposomes were used in coflotation and lipid binding assays. Vesicles used for proteoliposome reconstitution and lipid mixing assays were additionally passed 21 times through a 0.1 μm Nuclepore membrane (Cytiva). For the reconstitution of MSP2N2-based lipid nanodiscs, lipid films were prepared as described for liposome reconstitution, except omitting DOPE and

cholesterol from the lipid mixture (in the case of the 25 mol% DOPS/5 mol% PI(4,5)P$_2$ and 15 mol% DOPS/2 mol% PI(4,5)P$_2$ nanodiscs) or using lower DOPE and cholesterol ratios when assembling the 25 mol% DOPS/5 mol% PI(4,5)P$_2$/5 mol% cholesterol and 15 mol% DOPS/5 mol% cholesterol nanodiscs. For nanodisc reconstitution, the dried lipid films were hydrated in reconstitution buffer supplemented with 1.7% or 0.6% (w/v) DDM, resulting in final lipid concentrations of 13.2 mM or 5.6 mM, respectively, and briefly sonicated with the microtip (Branson Sonifier 250).

## Reconstitution of ferlins into proteoliposomes

Dysferlin and myoferlin (Fig. EV4D,E) were reconstituted into 100 nm LUVs by mixing DOPC/DOPE-only liposomes, OG (n-Octyl-β-D-glucopyranoside, Glycon Biochemicals) and the purified ferlin (in DDM micelles) at a protein-to-lipid ratio of 1:3500, an R-value of 1, and a final lipid concentration of 4 mM. The samples were then incubated for 20 min at room temperature and transferred to Slide-A-Lyzer MINI dialysis devices (MWCO 10 kDa, Thermo Fisher Scientific) or Slide-A-Lyzer cassettes (MWCO 10 kDa, Thermo Fisher Scientific) for overnight dialysis at 4–8 °C against 2 L of reconstitution buffer (20 mM HEPES-KOH, pH 7.5, 150 mM KCl), supplemented with 2.5 g/L Bio-Beads SM-2 (Bio-Rad). Bio-Beads SM-2 were prepared by sequential washing with methanol, ethanol, ddH$_2$O, and reconstitution buffer. The successful reconstitution of ferlins into liposomes was verified by liposome flotation on a Nycodenz step gradient (0%/30%/40%) following ultracentrifugation for 90 min at 50,000 r.p.m. in a TLS-55 rotor (Beckman Coulter).

## Liposome binding assays

To assess the ability of soluble myoferlin (1–1997) to interact with model lipid bilayers, 50 μM Dansyl-labelled LUVs (labelled with 5 mol% 18:1 Dansyl PE), of a varying anionic phospholipid composition, were mixed with 0.9 μM ferlin sample in the presence of Ca$^{2+}$. The total reaction volume was 15 μL. Following incubation for 5 min at room temperature, the protein-lipid samples were transferred into an ultra-micro cuvette (QS 105.252, 1.5 mm optical path, Hellma) and the emission spectra were taken between 450–560 nm at a 284 nm excitation wavelength (3 nm slits, 0.1 s integration time) using a Fluorolog 3 spectrofluorometer (Horiba Jobin Yvon). The relative protein-to-membrane FRET (Förster Resonance Energy Transfer) efficiency was calculated as follows: rFRET = $(I - I_{min})/(I_{max} - I_{min})$, where $I$ represents the average emission intensity of the sample at 518–520 nm, $I_{min}$—the intensity of the protein-free sample, and $I_{max}$—the maximum Dansyl emission of the titration series. To estimate the $[Ca^{2+}]_{1/2}$ values, the liposome binding data were fitted to the Hill equation (Brandt et al, 2012) in GraphPad Prism 10. All experiments were performed in triplicate ($n = 3$, technical replicates).

## Coflotation assays

In typical liposome coflotation experiment, soluble ferlin constructs were mixed with LUVs in presence of 50 μM (Appendix Fig. S1D; Fig. EV4F,G) or 0.5 mM (Appendix Fig. S12F) Ca$^{2+}$ or Mg$^{2+}$ and added to the bottom layer of a Nycodenz step gradient (0%/30%/40%). The final protein and liposomes assay concentrations were 1 μM and

1 mM, respectively. The bottom Nycodenz layer (40% (w/v)) was overlaid with equal volumes (40 µL) of a 30% (w/v) Nycodenz solution and reconstitution buffer (20 mM HEPES-KOH, pH 7.5, 150 mM KCl). The Nycodenz gradients were centrifuged at 50,000 r.p.m. for 90 min in a TLS-55 rotor (Beckman Coulter), harvested from the top in 20 µL fractions, and analysed by SDS-PAGE (NuPAGE 4–12% Bis-Tris gels, Thermo Fisher Scientific). The SDS-PAGE gels were stained with One-Step Blue (Biotium). Under these experimental conditions, liposomes float to the top two Nycodenz fractions and comigration of ferlins to the top fractions is indicative of their interaction with the lipid bilayer. All coflotation experiments were conducted at least three times (technical and biological replicates).

## Lipid mixing assays

In a typical "bulk" lipid mixing assay (Hernandez et al, 2012; Hoekstra and Duzgunes, 1993; Yavuz et al, 2018), used to monitor the ability of full-length myoferlin and dysferlin to promote tight vesicle-vesicle docking, 20 µL empty dual-labelled LUVs (100 nm, comprising 1 mol% NBD PE and 1 mol% Rhodamine B PE) were mixed in a 1:1 ratio with unlabelled proteoliposomes in 1 mL reconstitution buffer (20 mM HEPES-KOH, pH 7.5, 150 mM KCl), supplemented with 0.1–1 mM $CaCl_2$. The lipid mixing data (Fig. EV4D,E) were acquired at 37 °C using a Fluorolog 3 spectrofluorometer (Horiba Jobin Yvon) and corrected for signal intensity variations (the S/R acquisition mode). The extent of NBD (donor) dequenching because of lipid mixing was monitored at 460 nm (3 nm slit) and 538 nm (3 nm slit) excitation and emission wavelengths, respectively. The lipid mixing reactions were stopped upon addition of 5 µL 10% (v/v) Triton X-100 (in reconstitution buffer), and the NBD dequenching signal, after detergent solubilization of liposomes, was considered as the maximal fluorescence ($F_{max}$). The normalized lipid mixing efficiency was calculated as: $(F−F_i)/(F_{max}−F_i)$, where $F_i$ represents the initial fluorescence of the labelled liposomes and $F_{max}$—the final fluorescence of the sample (after detergent addition). The dual-labelled LUVs comprised 25 mol% DOPS and 5 mol% $PI(4,5)P_2$ anionic phospholipids, whereas full-length myoferlin and dysferlin were reconstituted in DOPC/DOPE-only LUVs. The control reactions included protein-free liposomes. The lipid mixing assays were repeated at least three times (technical replicates) and at least two separate proteoliposome reconstitutions (biological replicates). The lipid mixing data was analysed in OriginPro 2020 (v9.7).

## Mass photometry characterization of full-length human dysferlin

All measurements (Appendix Fig. S1H) were performed with the OneMP mass photometer (Refeyn). Images were acquired with Refeyn AcquireMP and analysed using Refeyn DiscoverMP software. For mass photometry measurements, twin-StrepII-tagged full-length dysferlin (residues 1–2080) was reconstituted into LMNG (Lauryl maltose neopentyl glycol, Anatrace) micelles and purified by anion-exchange chromatography. The dysferlin sample was concentrated to ~3 mg/mL in the presence of 0.01% (w/v) LMNG and dialysed before measurements using a Slide-A-Lyzer MINI device (MWCO 20 kDa, Thermo Fisher Scientific) against the dialysis buffer (20 mM HEPES-KOH, pH 7.5, 150 mM KCl, 2 mM DTT). The sample was then 20-fold diluted in the dialysis buffer and centrifuged at 13,000 r.p.m. for

10 min at 4 °C before the measurements. For each measurement, 1 µL of sample was added to a droplet of 12 µL of dialysis buffer. For the experiments carried out in the presence of $CaCl_2$, the sample was diluted in a dialysis buffer containing 2 mM or 4 mM $CaCl_2$ and incubated for 45–60 min, before performing the measurement. Mass calibration was achieved by adding 6 µL of NativeMark™ Unstained Protein Standard (Invitrogen) diluted 100-fold to a 12 µL drop of the dialysis buffer and using the peaks corresponding to bovine serum albumin (66 kDa), lactate dehydrogenase (146 kDa) and apo-ferritin (480 kDa). Each experiment was repeated four times (technical replicates).

## Cryo-EM sample preparation

To preserve the integrity of ferlin samples in vitreous ice, lipid-free soluble myoferlin (1–1997) and dysferlin (1–2017) were stabilized through glutaraldehyde (GA, Electron Microscopy Sciences) cross-linking during gradient centrifugation (GraFix) (Kastner et al, 2008). Prior to vitrification, 80 µL of soluble dysferlin (1–2017) at ~5.5 mg/mL was applied to a linear 5–40% (w/v) sucrose gradient prepared by mixing equal volumes of the light gradient (20 mM HEPES-KOH, pH 7.5, 150 mM KCl, 1 mM $CaCl_2$, 5% (w/v) sucrose) and GA-supplemented heavy gradient solution (20 mM HEPES-KOH, pH 7.5, 150 mM KCl, 1 mM $CaCl_2$, 40% (w/v) sucrose, 0.2% (v/v) GA) using the Gradient Master 108 (Biocomp). The gradient was centrifuged at 4 °C for 15 h at 29,100 r.p.m. in an SW40Ti rotor (Beckman Coulter). In the next step, the gradient was harvested from the top in 500 µL fractions using the Piston Gradient Fractionator (Biocomp), and the crosslinker was immediately quenched with 50 mM L-lysine and L-arginine (final concentration, f.c.). After incubation for 2 h on ice, the monomeric fractions were pooled and concentrated to ~80 µL, transferred to a Slide-A-Lyzer MINI device (Thermo Fisher Scientific, MWCO 10 kDa), and then dialyzed overnight against 2 L of the minimal buffer (20 mM HEPES-KOH, pH 7.5, 150 mM KCl, 1 mM $CaCl_2$, 1 mM DTT, 2.5% (v/v) glycerol). Following an additional 2 h dialysis against 1 L of minimal buffer, the concentration was adjusted to A280 ~ 1.08 (absorbance at 280 nm), and the sample was used directly for cryo-EM grid preparation. Cryo-EM grids were prepared using a Vitrobot Mark IV plunger (Thermo Fisher Scientific), operated at 4 °C and 100% humidity. Soluble dysferlin (1–2017) grids suitable for data collection were obtained by applying 3 µL of the cross-linked sample to one side of UltrAuFoil R1.2/1.3 300 gold mesh grids (Jena Bioscience), which were pretreated with the Plasma Cleaner (Harrick Plasma) for 1 min at medium settings before vitrification in liquid ethane, cooled by liquid nitrogen. For optimal sample vitrification, the cryo-EM grids were blotted for 2–3 s using a blot force of 5 and stored in liquid nitrogen prior to screening and data collection.

Like dysferlin, lipid-free myoferlin (1–1997) was stabilized through GraFix. Myoferlin (1–1997) samples were prepared in the minimal buffer containing 200 mM KCl (25 mM HEPES-KOH, pH 7.5, 200 mM KCl, 2.5% (v/v) glycerol, 1 mM TCEP) in the presence of 0.5 mM $CaCl_2$ and 50 µM WJ460 (GlpBio), and then centrifuged for 90 min at 50,000 r.p.m. on 5–40% (w/v) sucrose GraFix gradients using a TLS-55 rotor (Beckman Coulter). The lipid-free myoferlin (1–1997) gradients were harvested from the top in 100 µL fractions, and the crosslinker was quenched with 50 mM L-lysine and L-arginine (f.c.). The peak gradient fractions of

myoferlin (1–1997) were concentrated, and the buffer was exchanged for the minimal buffer (containing 0.5 mM $CaCl_2$) using a Vivaspin 500 concentrator (MWCO 50 kDa, Sartorius). Lipid-free myoferlin (1–1997) was vitrified in a mixture of liquid ethane and propane (37%:63%) cooled by liquid nitrogen, following the application of 3 μL sample at A280 ~ 1.15 to a plasma-treated Quantifoil R1.2/1.3 200 copper mesh grid (Jena Bioscience), which was blotted for 7.5 s using a blot force of 3.

To assemble lipid-bound myoferlin (1–1997) complexes (Appendix Fig. S1B,C), empty nanodiscs with the desired lipid composition were reconstituted in the presence of the MSP2N2 scaffold, as recently described (Cannon et al, 2023). Purified His-tagged human MSP2N2 (~2.9 mg/mL, obtained from Cube Biotech) and DDM-solubilized lipids were mixed at ~1:80–1:200 protein:lipid ratios, incubated at room temperature for 20 min, and dialyzed overnight at 4–8 °C against 2 L of reconstitution buffer (25 mM HEPES-KOH, pH 7.5, 150–200 mM KCl, 1 mM DTT), supplemented with 2.5 g/L Bio-Beads SM-2. Empty nanodiscs were subsequently purified by size-exclusion chromatography (SEC) on Superdex 200 Increase 10/300 GL (Cytiva), equilibrated in the SEC buffer (25 mM HEPES-KOH, pH 7.5, 150–200 mM KCl, 1.25–2.5% (v/v) glycerol, 1 mM DTT (or 0.5 mM TCEP), 0.5 mM $CaCl_2$). Peak MSP2N2 nanodisc fractions were concentrated to ~2.2 mg/mL and added in ~2-fold molar excess over purified soluble myoferlin (1–1997), followed by incubation for 30 min at room temperature. The formation of myoferlin (1–1997)-nanodisc complexes (Appendix Fig. S1B,C) was assessed by SEC on a Superose 6 Increase 10/300 GL (Cytiva) column, equilibrated in the SEC buffer. For cryo-EM grid preparation, the myoferlin-nanodisc complexes were cross-linked on ice in batch with 0.05–0.08% (v/v) GA for 10–30 min, and the reaction was stopped with 50 mM L-lysine and L-arginine (f.c.) for 15 min on ice. Following centrifugation at 14,800 r.p.m. at 4 °C for 5 min, the cross-linked complexes were applied to a Superose 6 column, and myoferlin (1–1997)-nanodisc fractions were pooled and concentrated by ultrafiltration to A280 ~ 0.7–0.8. Lipid-bound myoferlin (1–1997) complexes were vitrified in liquid ethane-propane (37%:63%) following the application of 3 μL of sample to plasma-cleaned Quantifoil R1.2/1.3 200 copper mesh grids, which were blotted for 6.5–8 s at a blot force of 3 (except for the 15% DOPS and 2 mol% PI(4,5)$P_2$ myoferlin (1–1997) nanodisc-myoferlin complex, frozen on an UltrAuFoil R1.2/1.3 300 gold mesh grid).

## Cryo-EM data collection and processing of lipid-bound myoferlin complexes

Sample size calculation was not performed, and no randomization or blinding was required. All lipid-bound myoferlin cryo-EM datasets were acquired on a Titan Krios G4 electron microscope (Collaborative Laboratory and User Facility for Electron Microscopy, Georg-August-Universität Göttingen), operated at an accelerating voltage of 300 kV and equipped with a Falcon4i direct electron detector and a Selectris X zero-loss energy filter (Appendix Figs. S2, S4 and S5, Tables S1–S4). All cryo-EM movies were recorded with EPU (Thermo Fisher Scientific) at 165,000x nominal magnification at the specimen level, resulting in a 0.72 Å/pixel exposure sampling rate. All datasets were collected using an energy filter slit width of 10 eV and a 50 μm C2 aperture (the objective aperture was not inserted). Cryo-EM movies were stored as raw camera frames (in EER format) following exposure over

~3.16–3.46 s, resulting in a total fluence of ~38.03–39.89 e⁻/Å² (Appendix Tables S1–S4). The EER movies were fractionated in 40 EER fractions during on-the-fly preprocessing (patch motion correction, patch CTF estimation, dose-weighting) with cryoSPARC Live (v.4.4-v.4.5) and curated (contaminated or low-resolution exposures were removed from subsequent analyses). Myoferlin particles were picked using the Blob and Template Picker (or with crYOLO (Wagner et al, 2019), Appendix Fig. S4A), extracted in 360 pixel boxes and 2x binned before being subjected to 3D and 2D classification in cryoSPARC. The initial particle sets were first cleaned by supervised 3D classification (Heterogeneous refinement) using one "good", ab initio generated reference volume and 4–5 "decoy" classes. Particles assigned to nanodisc-bound myoferlin classes were further classified in 2D, refined in 3D, and re-extracted in a 360 pixel box (0.72 Å/pixel) before Non-uniform (NU) refinement in cryoSPARC. To resolve the peripheral $C_2G$ and the missing inner DysF domain (from some particles), the cryoSPARC-refined particles were subjected to 3D classification in RELION-5.0-beta (Kimanius et al, 2024). Briefly, the myoferlin-nanodisc particles were first classified in 3D without image alignment using 4 classes, with soft masks applied to the inner DysF motif or to $C_2G$, the inner and outer DysF (Appendix Figs. S2C, S4A, S5C, and S5F). The subset of particles exhibiting stronger density for the inner DysF and $C_2G$ were then re-imported and 3D-refined in cryoSPARC (Appendix Figs. S2C, S4A, S5C, and S5F). Prior to focused classification, the 15 mol% DOPS and 5 mol% cholesterol nanodisc-myoferlin particles were subjected to an additional round of global 3D classification into 5 classes (Appendix Fig. S5C). For all lipid-bound myoferlin complexes, the final particle sets were subjected to sequential CTF refinement and reference-based motion correction in cryoSPARC to obtain the final maps. To facilitate model building, the lipid-bound myoferlin maps were sharpened using LocScale (Jakobi et al, 2017) and DeepEMhancer (Sanchez-Garcia et al, 2021), and the local resolution of the maps was estimated in cryoSPARC. The continuous flexibility and conformational space of membrane-bound myoferlin particles were evaluated in cryoSPARC using the 3D variability analysis (3DVA) tool (Movie EV2).

## Cryo-EM data collection and image analysis of lipid-free myoferlin and dysferlin datasets

The lipid-free myoferlin and dysferlin datasets were acquired using EPU on the same microscope and detector (Falcon4i/Selectris), with the energy filter slit width set to 10 eV or 15 eV (Fig. EV3A; Appendix Fig. S6A, and Tables S5–S6). Lipid-free dysferlin (1–2017) cryo-EM movies were recorded at a nominal magnification of 165,000× from the same grid in two sessions over a ~3.0 s exposure, resulting in total fluences of ~40.18 e⁻/Å² and ~40.22 e⁻/Å² (Appendix Fig. S6A and Table S6). Cryo-EM movies of lipid-free myoferlin (1–1997) were collected in two sessions at total fluences of ~39.91 e⁻/Å² and ~39.85 e⁻/Å², respectively (Fig. EV3A; Appendix Table S5). Lipid-free ferlin datasets were pre-processed and curated on-the-fly in cryoSPARC Live (v.4.1 for lipid-free dysferlin, v.4.4-v.4.5 for lipid-free myoferlin), including beam-induced motion correction, dose-weighting, patch CTF estimation, and particle picking (using the Blob and Template Picker). Particles were extracted in a 360 pixel box (0.72 Å/pixel), 2x binned, split into subsets of ~0.7–1 million particles (Fig. EV3A; Appendix Fig. S6A),

before supervised 3D classification in cryoSPARC with 4 or 5 classes, including an ab initio generated ferlin reference volume. After additional 2D classification, the lipid-free myoferlin and dysferlin particles were refined in 3D to obtain the consensus maps, where the Fer$^{core}$ region exhibited strong density. The C$_2$E and C$_2$F domains could also be resolved in the lipid-free myoferlin consensus maps (Fig. EV3A). As C$_2$B and the C-terminal C$_2$F-C$_2$G appeared dynamic in the imaged lipid-free ferlin particles, multiple rounds of global and focused 3D classifications were employed to identify discrete conformations of the domains. Consequently, focused classification with a soft mask applied to the C$_2$F and C$_2$G regions of the map allowed us to identity more homogeneous dysferlin particles; these particles refined in 3D to 3.5 Å (dysferlin (1–2017) map M2) and 4.8 Å (dysferlin (1–2017) map M3) resolutions. In these maps, C$_2$F appeared to establish a defined interface with the N-terminal C$_2$B, whereas C$_2$G was positioned near C$_2$F. By employing a similar 3D classification approach, we could resolve the C$_2$B-C$_2$F-C$_2$G interface in a subset of lipid-free myoferlin particles (Fig. EV3A). These particles refined in 3D to 3.2 Å (myoferlin (1–1997) map M9) and 8.8 Å (myoferlin (1–1997) map M11) global resolutions. Attempts to locate the N-terminal C$_2$A domain by masked 3D classification were not successful, consistent with its lack of significant contact interfaces with the remaining C$_2$ domains or with it being destabilized in vitreous ice (Appendix Fig. S14D).

## Model building and refinement

Model building was initiated using AlphaFold2 predictions of human myoferlin and dysferlin (Jumper et al, 2021; Mirdita et al, 2022). The theoretical models were prepared in Phenix (Oeffner et al, 2022), divided into individual domains (or structural modules, such as the Fer$^{core}$), fitted into the overall ferlin maps in ChimeraX (v.1.6-1.8) and UCSF Chimera (v.1.16), and further adjusted with ISOLDE (Croll, 2018; Oeffner et al, 2022). All structural models were manually corrected and rebuilt in Coot (v.0.9.8.5) (Emsley and Cowtan, 2004) and iteratively refined using *phenix.real_space_refine* (Phenix v.1.19.2-4158-1.21.2-5419) (Afonine et al, 2018), with *nonbonded_weight* set to 1000 for the final refinements.

Modelling of bound Ca$^{2+}$ ions and phospholipid headgroups was guided by $F_o$–$F_c$ omit maps, calculated with Servalcat (Yamashita et al, 2021). Geometry restraints for the refinement of the two resolved phosphatidylserine molecules were generated with Grade2 (https://grade.globalphasing.org). No cryo-EM density was observed for the WJ460 ligand (Zhang et al, 2018), added to both the lipid-free myoferlin (1–1997) and the 15 mol% DOPS and 2 mol% PI(4,5)P$_2$ nanodisc-myoferlin complex. Since the proposed binding site residues of C$_2$D are not accessible in the lipid-free and lipid-bound myoferlin cryo-EM structures, further studies are needed to clarify whether myoferlin is indeed the cellular target of this small-molecule compound (Zhang et al, 2018). All structural models were validated using MolProbity in Phenix. Structural figures were prepared with ChimeraX (v.1.6-1.8) and Pymol (v.2.5.4, Schrödinger LLC). Contact interface areas were estimated with PISA (v.1.5.2) (Krissinel and Henrick, 2007), and the centroid distances and angles between the rearranged myoferlin domains were estimated in UCSF Chimera (v.1.16). Data collection and refinement statistics are provided in Appendix Tables S1–S6.

## Chemical cross-linking mass spectrometry characterization of human dysferlin

Prior to mass spectrometry analysis (Appendix Fig. S14A–D), soluble dysferlin (1–2017) was complexed with 1.5 mM CaCl$_2$ and cross-linked in batch with 0.3 mM BS3 (Thermo Fisher Scientific) for 30 min at room temperature. The cross-linking reaction was stopped with 50 mM Tris-HCl, pH 8.0, and the sample was further purified by size-exclusion chromatography on a Superose 6 Increase 10/300 GL column (Cytiva), equilibrated in the sample buffer (20 mM HEPES-KOH, pH 7.5, 150 mM KCl, 5% (v/v) Glycerol, 2 mM DTT, 1.5 mM CaCl$_2$). Individual monomeric peak fractions (fraction 14 and 15) were ethanol precipitated and resuspended in 12.5 μl 8 M Urea. The samples were then diluted to a final concentration of 2 M Urea and reduced by the addition of 10 μl 50 mM DTT for 30 min at 37 °C. Alkylation was achieved by the addition of 10 μl 200 mM iodoacetamide and incubation for 30 min at 25 °C. Unreacted iodoacetamide was quenched with an additional 10 μl of 50 mM DTT. Protein digestion was performed overnight at 37 °C with 1 μg of trypsin (Promega) in the presence of 1 M Urea and 50 mM Tris-HCl, pH 7.75. The samples were acidified with formic acid to a final concentration of 0.1% (v/v) and acetonitrile was added to 5% (v/v) final concentration. Digested peptides were desalted with C18 reversed-phase MicroSpin columns (Harvard Apparatus). Bound peptides were eluted with 50% (v/v) acetonitrile, 0.1% (v/v) formic acid, dried under vacuum, resuspended in 75 μl 2% (v/v) acetonitrile, 0.05% (v/v) TFA, and 5 μl were used for LC-MS analysis. Chromatographic separation was achieved with Dionex Ultimate 3000 UHPLC (Thermo Fischer Scientific) coupled to an in-house packed C18 column (ReproSil-Pur 120 C18-AQ, 1.9 μm particle size, 75 μm inner diameter, 33 cm length, Dr. Maisch GmbH) over a 74 min linear gradient from 8 to 46% mobile phase B (mobile phase A: 0.1% (v/v) FA, mobile phase B: 80% (v/v) ACN, 0.08% (v/v) FA). Eluting peptides were analysed using Orbitrap Exploris 480 (Thermo Fischer Scientific) with the following settings for survey scans: resolution—120,000; scan range—380–1600; AGC target—300%; maximum injection time set to "Auto". Analytes with charge states of 3 to 8 were selected for fragmentation with 28% normalized collision energy. Dynamic exclusion was set to 15 s. Fragment spectra were acquired with the following settings: resolution—30,000; isolation window—1.6 *m/z*; AGC target—100%; maximum injection time—128 ms. Resulting.raw files were analysed with pLink (Chen et al, 2019) (v.2.3.9) against a database containing the protein sequence. Carbamidomethyl on cysteines was set as a fixed modification, and oxidation of methionines as a variable modification. BS3 was selected as the crosslinker, the peptide tolerance was set to 6 ppm, and the False discovery rate (FDR) was set to 1%. The identified cross-linked residues (Dataset EV1) were mapped onto the AlphaFold2 prediction of dysferlin (Appendix Fig. S15B) and the lipid-free dysferlin (1–2017) cryo-EM model using xiNet (Combe et al, 2015). Chemical cross-linking data were analysed and visualised in PyMOL (v.2.5.4).

## Data availability

The atomic coordinates and cryo-EM maps of the determined structures were deposited in the Protein Data Bank (PDB,

https://www.rcsb.org) and the Electron Microscopy Data Bank (EMDB, https://www.ebi.ac.uk/emdb) under the following accession codes: PDB 9H6X/EMD-51902 (2.56 Å, human myoferlin (1–1997)-nanodisc complex, 25 mol% DOPS and 5 mol% PI(4,5)P$_2$), 9qlf/EMD-53226 (2.65 Å, human myoferlin (1–1997)-nanodisc complex, 25 mol% DOPS, 5 mol% PI(4,5)P$_2$ and 5 mol% cholesterol), 9qkv/EMD-53222 (2.74 Å, human myoferlin (1–1997)-nanodisc complex, 15 mol% DOPS and 5 mol% cholesterol), 9qle/EMD-53225 (2.79 Å, human myoferlin (1–1997)-nanodisc complex, 15 mol% DOPS and 2 mol% PI(4,5)P$_2$), 9qln/EMD-53229 (3.21 Å, Ca$^{2+}$-bound lipid-free myoferlin (1–1997)), and 9qls/EMD-53233 (3.54 Å, Ca$^{2+}$-bound lipid-free dysferlin (1–2017)).

The source data of this paper are collected in the following database record: biostudies:S-SCDT-10_1038-S44318-025-00463-8.

# Peer review information

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

## Acknowledgements

We thank Christiane Senger-Freitag, Sandra Gerke, Sina Langer, Olivia Langer, Andy Schacht, and Patricia Räke-Kügler for excellent technical and administrative support. We are grateful to Dr. Tat Cheung Cheng and Prof. Dr. Ruben Fernandez-Busnadiego for supporting cryo-EM data acquisition and Dr. Eri Sakata for critically reading the manuscript. This work was supported and funded by Deutsche Forschungsgemeinschaft (DFG) through CR 937/2-1 (to CC) and SFB889 (to JP and TM), and the Cluster of Excellence (EXC2067) Multiscale Bioimaging EXC 2067/1-390729940 (CC, TM, and JP). CZKS and VP were supported by an Wellcome Trust Investigator grant (220300Z/20/Z). HU was supported by the Max Planck Society. Cryo-EM instrumentation was jointly funded by the DFG Major Research Instrumentation program (448415290) and the Ministry of Science and Culture of the State of Lower Saxony (Niedersächsisches Ministerium für Wissenschaft und Kultur). TM was also supported by the Leibniz Program of the DFG (MO896/5), the Ernst Jung Prize for Medicine, and by Fondation Pour l'Audition (FPA RD-2020-10).

## Author contributions

**Constantin Cretu**: Conceptualization; Data curation; Formal analysis; Funding acquisition; Validation; Investigation; Visualization; Methodology; Writing—original draft; Project administration; Writing—review and editing. **Aleksandar Chernev**: Resources; Data curation; Investigation; Methodology. **Csaba Zoltan Kibedi Szabo**: Resources; Data curation; Formal analysis; Investigation; Methodology. **Vladimir Pena**: Resources; Supervision; Funding acquisition; Writing—review and editing. **Henning Urlaub**: Resources; Supervision; Funding acquisition; Methodology. **Tobias Moser**: Conceptualization; Resources; Supervision; Funding acquisition; Project administration; Writing—review and editing. **Julia Preobraschenski**: Conceptualization; Supervision; Funding acquisition; Project administration; Writing—review and editing.

Source data underlying figure panels in this paper may have individual authorship assigned. Where available, figure panel/source data authorship is listed in the following database record: biostudies:S-SCDT-10_1038-S44318-025-00463-8.

## Funding

## Disclosure and competing interests statement

The authors declare no competing interests.

# Expanded View Figures

**Figure EV1. Cryo-EM density snapshots of lipid-bound soluble myoferlin (1–1997).**

(A) Selected cryo-EM density snapshots of $C_2B$ and its proximal motifs (the linker helix and the FerI motif). Myoferlin's residues are depicted as sticks. (B) Cryo-EM density of myoferlin's $C_2C$ domain. The three bound $Ca^{2+}$-ions and the recruited phosphatidylserine (PS) are indicated. (C) $Ca^{2+}$-binding sites of the $C_2C$ domain of myoferlin. The cryo-EM density is contoured around the $Ca^{2+}$-binding residues of $C_2C$. (D) Cryo-EM density of the FerA motif of myoferlin. Notably, the four-helix bundle domain inserts between the β4 and β5 strands of the $C_2CD$ domain (red, shown in a cartoon representation). (E) Cryo-EM density of the $C_2CD$ domain of myoferlin. The seven β-strands of the domain are indicated. (F) Cryo-EM densities of the modelled inner and outer DysF motifs of myoferlin. (G) Cryo-EM density and the observed $Ca^{2+}$-binding sites of the $C_2D$ domain of myoferlin. $C_2D$'s cryo-EM density is contoured around the $Ca^{2+}$-binding sites (bottom panel). (H) Density of the $C_2E$ domain. The domain comprises an extended insertion loop (residues 1395–1498, denoted as the anchor loop), inserted between the β6-β7 strands. The loop establishes contacts with the downstream $C_2F$ domain. (I) Cryo-EM density of $C_2F$ and its $Ca^{2+}$- and phospholipid-binding sites. Three $Ca^{2+}$ ions and a PS headgroup were identified in the cryo-EM density map. (J) Cryo-EM density of the C-terminal $C_2G$ domain, derived from the myoferlin (1–1997)-nanodisc complex containing 25 mol% DOPS and 5 mol% PI(4,5)$P_2$. The $C_2G$ domain is shown in two orientations, with its lipid-binding β-hairpin motif coloured light orange. (K) Density of the C-terminal $C_2G$ domain, resolved in the myoferlin (1–1997)-nanodisc complex containing 15 mol% DOPS and 2 mol% PI(4,5)$P_2$. Notably, the insertion loop of $C_2G$ (residues 1906–1944) is ordered in this myoferlin complex. (L) $Ca^{2+}$-binding sites of $C_2G$, observed in the lipid-bound myoferlin structure (15 mol% DOPS and 2 mol% PI(4,5)$P_2$ nanodisc). Due to the lower local resolution, $Ca^{2+}$ modelling was initiated by AlphaFold3 predictions (Abramson et al, 2024), and the optimal sites were refined against the cryo-EM map of the complex (Appendix Fig. S5F). Our modelling (right panel) suggests that two $Ca^{2+}$ ions are coordinated by the L1 and L3 loops of $C_2G$ (Corbalan-Garcia and Gomez-Fernandez, 2014; Rizo and Sudhof, 1998). (M) Cryo-EM densities of the $Ca^{2+}$-bound phosphatidylserine (PS), resolved in the myoferlin-nanodisc complexes. The cryo-EM density is contoured around the PS headgroups, and the ligands are depicted as sticks.

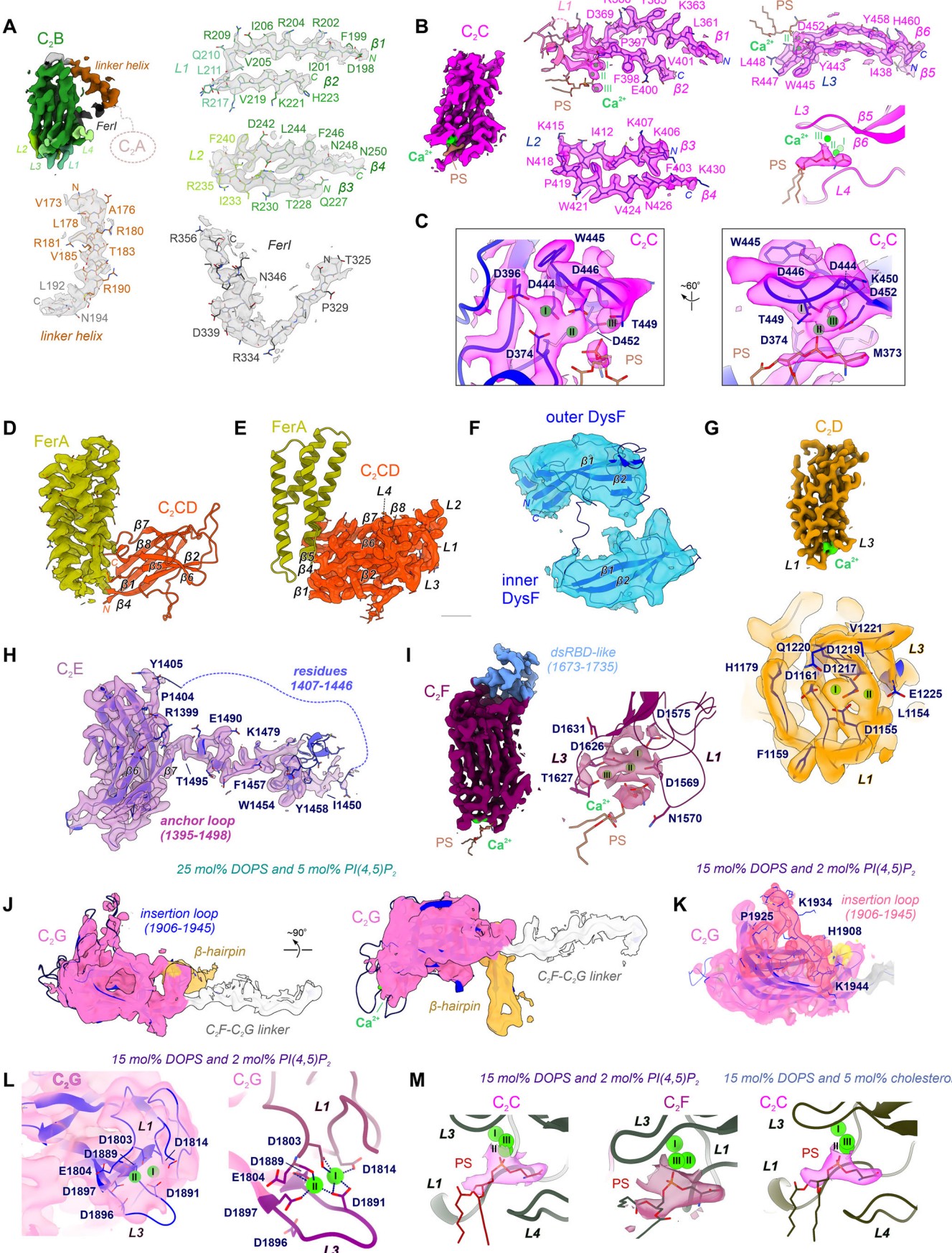

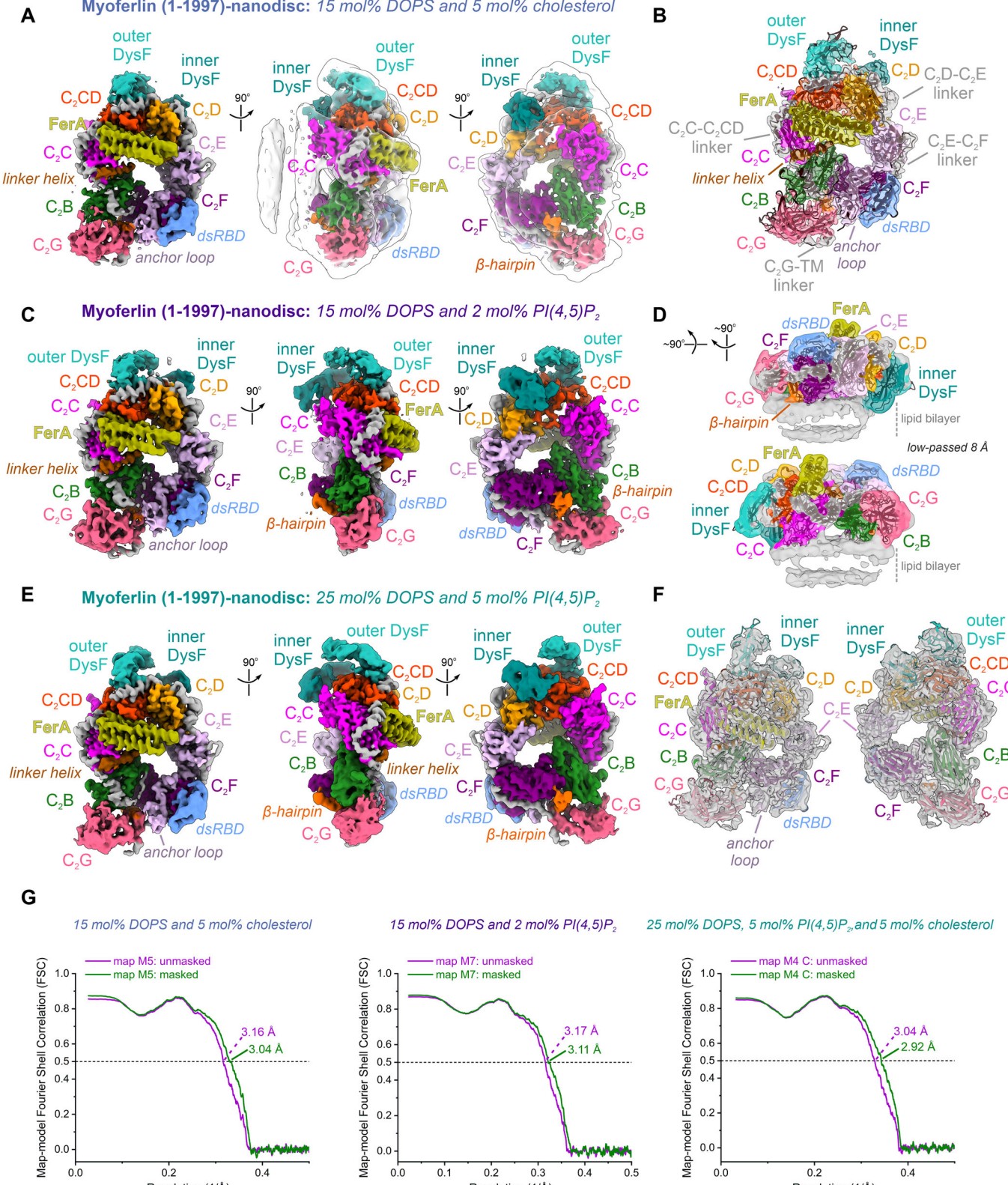

**A** Myoferlin (1-1997)-nanodisc: *15 mol% DOPS and 5 mol% cholesterol*

**B**

**C** Myoferlin (1-1997)-nanodisc: *15 mol% DOPS and 2 mol% PI(4,5)P₂*

**D**

**E** Myoferlin (1-1997)-nanodisc: *25 mol% DOPS and 5 mol% PI(4,5)P₂*

**F**

**G**

*15 mol% DOPS and 5 mol% cholesterol*

*15 mol% DOPS and 2 mol% PI(4,5)P₂*

*25 mol% DOPS, 5 mol% PI(4,5)P₂, and 5 mol% cholesterol*

◀ **Figure EV2.   Structural comparison between the different lipid-bound myoferlin complexes.**

(A) Overall cryo-EM map (map M6, ~3.3 Å) of soluble myoferlin (1–1997) bound to an MSP2N2 nanodisc containing 15 mol% DOPS and 5 mol% cholesterol (Appendix Fig. S5C). The map is shown in three different orientations. A lowpass filtered map (transparent surface) is superimposed to better visualize the nanodisc density. (B) Cryo-EM map of the myoferlin (1–1997)-nanodisc complex (15 mol% DOPS and 5 mol% cholesterol nanodisc). The final model is fitted inside the map (Appendix Fig. S5C). Several $C_2$ domain linker regions, the linker helix, and the anchor loop of $C_2E$ are well resolved in this myoferlin complex. (C) Overall cryo-EM map (map M8, ~3.43 Å) of the soluble myoferlin (1–1997) bound to a nanodisc containing 15 mol% DOPS and 2 mol% PI(4,5)P$_2$ (Appendix Fig. S5F). The map is shown in three different orientations, as in (A). (D) Side views of the myoferlin (1–1997)-nanodisc complex (map M8, 15 mol% DOPS and 2 mol% PI(4,5)P$_2$), together with the fitted model (Appendix Fig. S5F). The map was lowpass filtered to 8 Å, and the nanodisc density is indicated. (E) Cryo-EM map of the myoferlin (1-1997)-nanodisc complex assembled onto a 25 mol% DOPS and 5 mol% PI(4,5)P$_2$ MSP2N2 nanodisc (~2.56 Å, map M3, Appendix S2C). (F) Overall map of the nanodisc-bound myoferlin (1-1997) complex (map M3, 25 mol% DOPS and 5 mol% PI(4,5)P$_2$) with the fitted final model. Except for the flexible N-terminal $C_2A$, all myoferlin domains ($C_2B$-$C_2G$) and accessory motifs (FerA, the linker helix, the anchor loop) were accurately built, resulting in the near-complete structure of ferlin's cytosolic region (Appendix Figs. S2C and S3). (G) Map versus model Fourier Shell Correlation (FSC) plots for the myoferlin (1–1997)-nanodisc complexes (Appendix Figs. S4A and S5C,F).

 

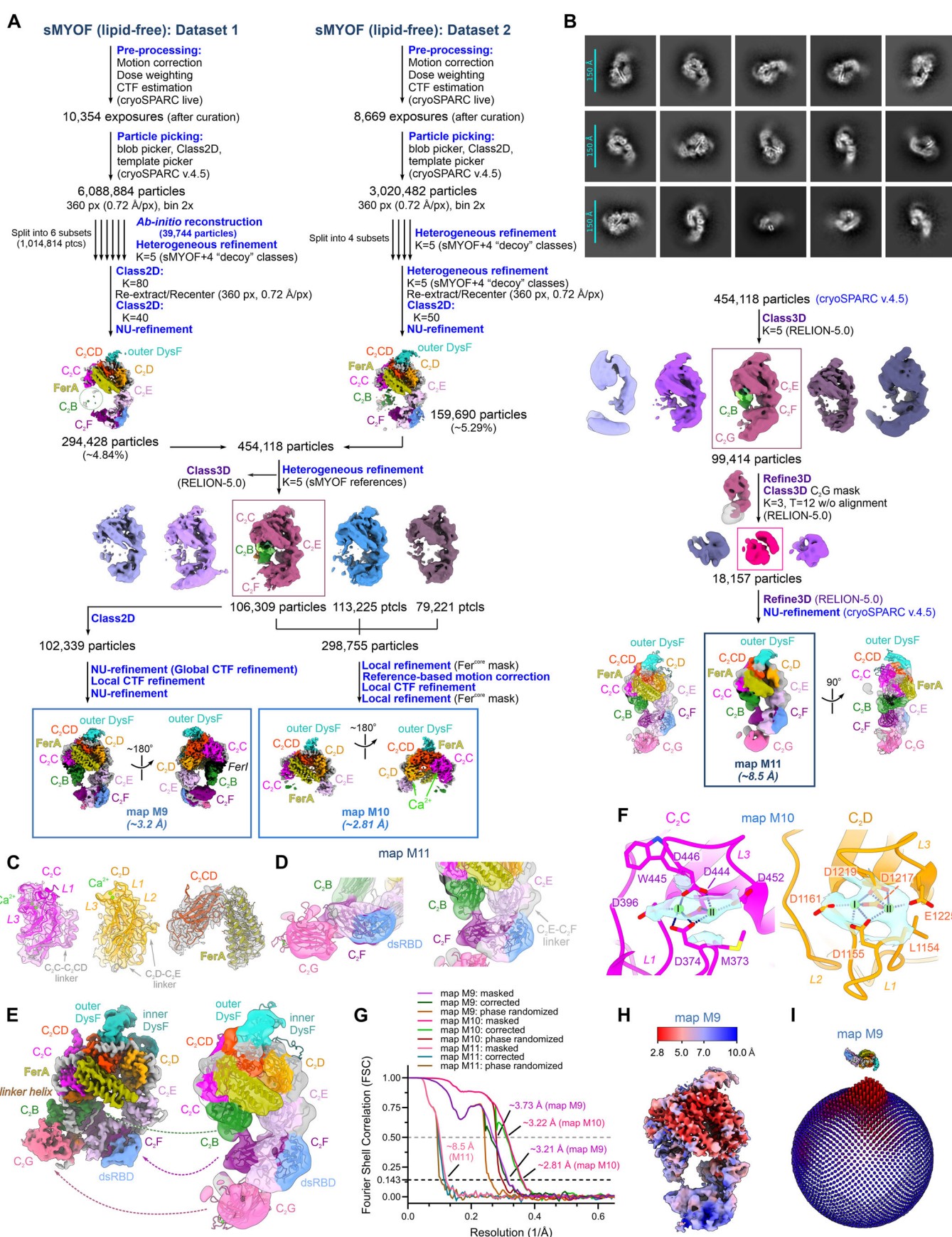

◀  **Figure EV3.  Cryo-EM image analysis of the vitrified lipid-free soluble myoferlin (1–1997).**

(A) Cryo-EM image processing schematic for the $Ca^{2+}$-bound, lipid-free myoferlin (1–1997). The resolution of the final maps (M9, M10, and M11 maps) was estimated according to the gold-standard Fourier Shell Correlation (FSC) criterion of 0.143. The structural domains of myoferlin are colour-coded as in Fig. 1. (B) Reference-free 2D class averages of lipid-free myoferlin (1–1997). Note the similarities between the dysferlin (1–2017) (Appendix Fig. S6C) and myoferlin (1–1997) 2D class averages in their lipid-free states. (C) Cryo-EM density of myoferlin's $C_2C$, $C_2D$, and $C_2CD$-FerA domains. The myoferlin model is fitted inside, and the $Fer^{core}$ map of myoferlin (map M10) is shown as a transparent surface. (D) Cryo-EM density of the $C_2F$ (map M9) and $C_2G$ (map M11) domains as modelled in the lipid-free myoferlin (1–1997) structure. The tertiary interfaces between the $C_2F$-$C_2B$ and $C_2F$-$C_2G$ are also observed in the lipid-free dysferlin (1–2017) structure (Appendix Figs. S6A, S7G–J, and S8H,I). (E) Side-by-side comparison between the membrane-bound (map M3) and lipid-free (map M11) myoferlin structures. The domains undergoing significant displacement upon nanodisc binding ($C_2B$, $C_2F$, and $C_2G$) are indicated with dashed arrows. (F) $Ca^{2+}$-binding sites observed in the lipid-free myoferlin structure. The cryo-EM density is coloured cyan and shown as a transparent surface. The two modelled $Ca^{2+}$ ions, bound to $C_2C$ and $C_2D$, are coloured green. (G) Global resolution estimates for the lipid-free myoferlin (1–1997) cryo-EM maps using Fourier Shell Correction (FSC) between half-maps. The resolution estimates at the FSC = 0.143 and FSC = 0.5 thresholds are indicated. (H) Local resolution of the overall map of the lipid-free soluble myoferlin (map M9). The map regions coloured in red indicate higher resolution. (I) Angular distribution of the myoferlin particles contributing to the overall map (map M9). The final cryo-EM map is shown above the 3D angular distribution representation. Relative cylinder height and the red colour indicate a higher number of particle images. Source data are available online for this figure.

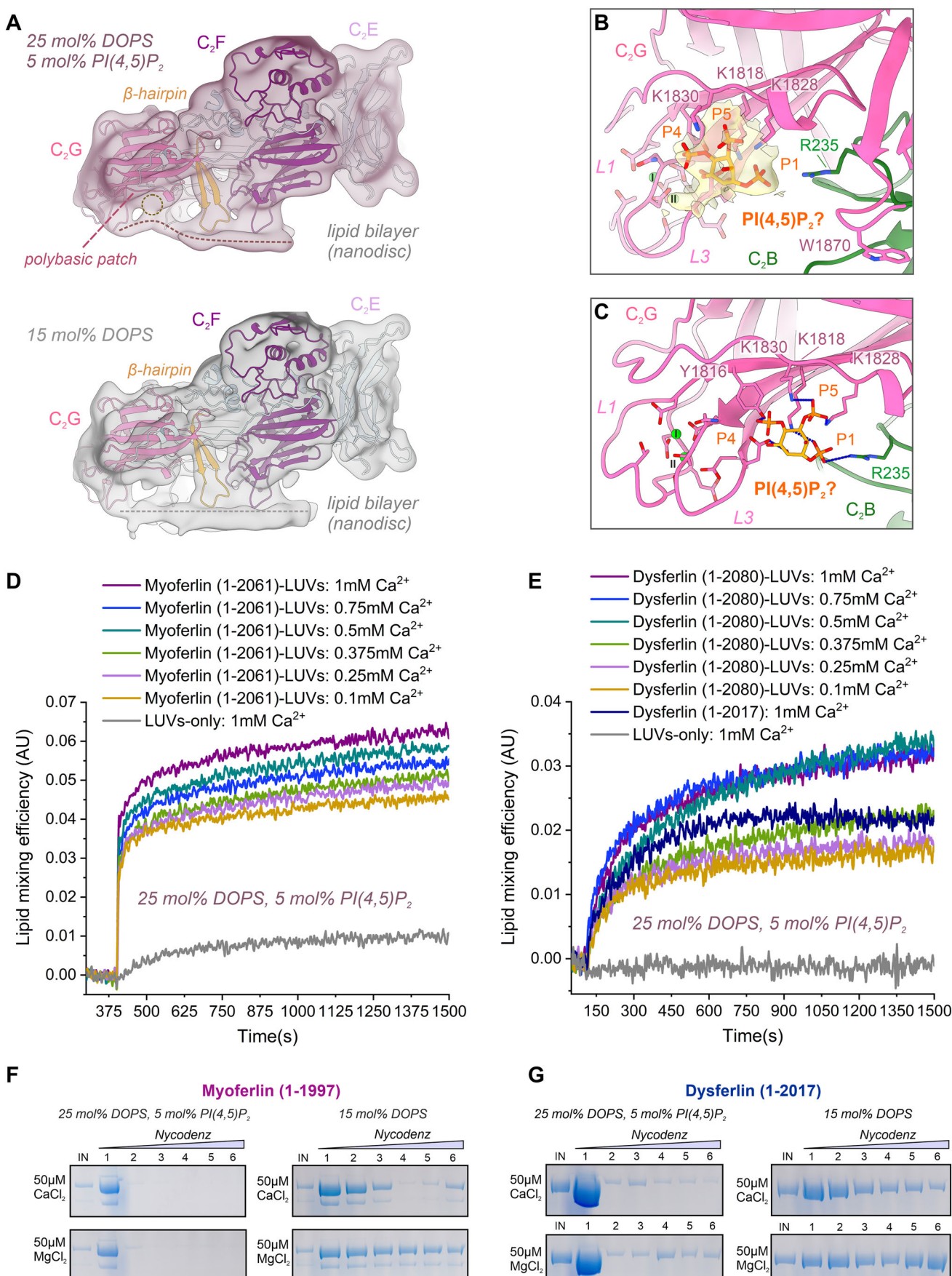

◀

**Figure EV4.  Full-length myoferlin and dysferlin appear to promote tight binding between vesicles.**

(A) Comparison of the local nanodisc structures observed in the myoferlin (1–1997)-nanodisc complexes. The lipid nanodisc forms close contacts with the concave surface of $C_2G$ in the presence (top, 25 mol% DOPS, 5 mol% PI(4,5)$P_2$, and 5 mol% cholesterol nanodisc, Appendix Fig. S4A–E), but not in the absence of PI(4,5)$P_2$ (bottom, 15 mol% DOPS and 5 mol% cholesterol nanodisc, Appendix Fig. S5C). The cryo-EM maps have been lowpass filtered to 10 Å, and the myoferlin model is fitted inside. (B) The cryo-EM density element proximal to $C_2G$ could accommodate a PI(4,5)$P_2$ headgroup. $C_2G$ and $C_2B$ residues located at the interface are displayed as sticks. The putative PI(4,5)$P_2$ density lies in close proximity to three lysine residues (K1818, K1828, and K1830), projecting from the concave surface of $C_2G$, as well as to an arginine residue of $C_2B$ (R235). The $Ca^{2+}$-binding L3 loop of $C_2G$ is also in close proximity. (C) Polar contacts between several basic residues of $C_2G$ and $C_2B$ and a tentative phospholipid, possibly PI(4,5)$P_2$, originating from the nanodisc bilayer. The nanodisc contained 25 mol% DOPS and 5 mol% PI(4,5)$P_2$. (D) Lipid mixing assays between myoferlin proteoliposomes and PS/PI(4,5)$P_2$-bearing vesicles in the presence of varying $Ca^{2+}$ concentrations. The fluorescent liposomes (Large unilamellar vesicles (LUVs), 25 mol% DOPS and 5 mol % PI(4,5)$P_2$) were dual-labelled with the Lissamine Rhodamine B and NBD (Nitrobenzoxadiazole). The NBD dequenching signal was used to monitor the extent of tight vesicle-vesicle docking (and, possibly, fusion) to non-fluorescent proteoliposomes. LUVs lacking full-length myoferlin (1–2061) were used as a control. Fluorescence traces were smoothed (using the Savitzky-Golay method) and normalized to the maximal dequenching signal to calculate the lipid mixing efficiency. The initial fluorescence increase is likely due to ferlin-induced aggregation of proteoliposomes. The lipid mixing assays were performed in triplicate, and representative fluorescence traces are shown. (E) Lipid mixing assays between dysferlin proteoliposomes and PS/PI(4,5)$P_2$-containing vesicles. The ability of full-length dysferlin (1–2080) to promote tight vesicle-vesicle docking was assessed as a function of $Ca^{2+}$ concentration. The lipid mixing activity of soluble dysferlin (1–2017) was also tested (deep blue). Empty LUVs were used as a control. All assays were repeated at least three times, and representative fluorescence traces are shown. (F, G) $Ca^{2+}$-dependent liposome binding activity of soluble myoferlin (1–1997) and dysferlin (1–2017) assessed using a coflotation assay. The liposomes used in these assays had a similar lipid composition to those in (D, E). The Nycodenz step gradients (0%/30%/40%) were harvested from the top and analysed by SDS-PAGE. Both myoferlin (1–1997) and dysferlin (1–2017) showed increased binding to LUVs containing both DOPS and PI(4,5)$P_2$. Source data are available online for this figure.

