## [Peer Review File · The EMBO Journal]

Structural insights into lipid membrane binding by human ferlins

Constantin Cretu, Aleksandar Chernev, Csaba Szabo, Vladimir Pena, Henning Urlaub, Tobias Moser, and Julia Preobraschenski

Corresponding author(s): Tobias Moser (tmoser@gwdg.de) , Julia Preobraschenski (julia.preobraschenski@med.uni-goettingen.de), Constantin Cretu (constantin.cretu@med.uni-goettingen.de)

Review Timeline:

Submission Date:	26th Oct 24
Editorial Decision:	3rd Dec 24
Revision Received:	24th Jan 25
Accepted:	25th Apr 25

Editor: Ioannis Papaioannou

Transaction Report:

Dear Tobias,

Thank you again for submitting your manuscript EMBOJ-2024-119411 for consideration by The EMBO Journal. As I have already informed you, it has now been seen by two experts in the field, and we have received their comments, which are included below.

Both referees find this work a significant contribution to the mechanistic understanding of how ferlins function, they commend the data for their high quality and solidity, mention that the experiments are properly controlled, and the manuscript well-written. Referee #1 supports publication of the manuscript as is, while referee #2 lists a few suggestions for improving the clarity of the manuscript further.

Given the referees' very supportive comments and recommendations, I would like to invite you to submit a revised version of your manuscript along with a detailed point-by-point response addressing all referees' comments. No further experimental work will be required for the revised manuscript to be accepted for publication in The EMBO Journal, but we kindly request you to sufficiently address all referees' comments by improving the text and figures as necessary.

I should also note that, as a matter of policy, any competing manuscripts published during review and revision of your manuscript will not negatively impact our assessment of the conceptual advance presented by your study. However, we request that you cite and discuss any related work in your revised manuscript.

Please let me know if you have any questions or comments that you would like to discuss with me. Thank you again for the opportunity to consider your work for publication in The EMBO Journal. I look forward to your revision.

Best regards,

Ioannis

Instructions for preparing your revised manuscript

1. When you are ready to submit the revision, please upload:

- A Word file of the manuscript text (including legends of main Figures, EV Figures and Tables). Please make sure that changes are highlighted (or "tracked") to be clearly visible.

- Individual production-quality figure files (one file per figure). When assembling your figures, please refer to our figure preparation guidelines in order to ensure proper formatting and readability in print as well as on screen:

If the data shown in a figure are obtained from n {less than or equal to} 2, please use scatter plots showing the individual data points.

- i. the name of the statistical test used to generate error bars and P values
- ii. the number (n) of independent experiments (please specify technical or biological replicates) underlying each data point (discussion of statistical methodology can be reported in the Materials and Methods section, but figure legends should contain a basic description of n , P , and the test applied)
- iii. the nature of the bars and error bars (s.d., s.e.m.).

- A point-by-point response to the referees' comments, with a detailed description of the changes made (as a word file). All referees' concerns must be fully addressed and their suggestions taken on board. When preparing your letter of response to the referees' comments, please bear in mind that this will form part of the Review Process File and will therefore be available online to the community. Please note that you have the possibility to opt out of the transparent process at any stage prior to publication by letting the editorial office know (contact@embojournal.org); if you do opt out, the Review Process File link will point to the

following statement: "No Review Process File is available with this article, as the authors have chosen not to make the review process public in this case.". For more details on our Transparent Editorial Process, please visit our website: <https://www.embopress.org/page/journal/14602075/authorguide#transparentprocess>

- Expanded View (EV) files (replacing Supplementary Information) that are collapsible/expandable online. A maximum of 5 EV Figures can be typeset. EV Figures should be cited as "Figure EV1, Figure EV2" etc. in the text, and their respective legends should be included in the manuscript file after the legends of regular figures. See detailed instructions regarding Expanded View files here:

- For the figures that you do NOT wish to display as Expanded View figures, they should be bundled together with their legends in a single PDF file called "Appendix", which should start with a short Table of Contents (including page numbers). Appendix figures should be referred to in the main text as: "Appendix Figure S1, Appendix Figure S2" etc. Please see detailed instructions here: <https://www.embopress.org/page/journal/14602075/authorguide#expandedview>

- A complete author checklist, which you can download from our author guidelines (<https://www.embopress.org/page/journal/14602075/authorguide>). Please note that the checklist will also be part of the Review Process File.

2. Please note that no statistics should be calculated and shown in Figures if $n=2$. Please also note that each p value should be reported as an exact value.

3. Before submitting your revision, primary datasets (and computer code, where appropriate) produced in this study need to be deposited in appropriate public databases (see <https://www.embopress.org/page/journal/14602075/authorguide#dataavailability>).

In particular, we kindly request you to deposit the atomic coordinates, cryo-EM, and mass spectrometry data produced in your study to appropriate databases. The accession numbers, databases, and the specific URLs (links) should be listed in a formal "Data availability" statement (placed after Methods).

*** All links should resolve to a page where the data can be publicly accessed. ***

4. Please check that the title and the abstract of the manuscript are brief, yet explicit, even to non-specialists. The length of the title should not exceed 100 characters, and the abstract should be a single paragraph not exceeding 175 words.

5. Please also note our reference format: <https://www.embopress.org/page/journal/14602075/authorguide#referencesformat>.

7. Please remember: digital image enhancement is acceptable practice, as long as it accurately represents the original data and conforms to community standards. If a figure has been subjected to significant electronic manipulation, this must be noted in the figure legend or in the "Materials and Methods" section. The editors reserve the right to request original versions of figures and the original images that were used to assemble the figure.

8. Our journal encourages inclusion of data citations in the reference list to directly cite datasets that were obtained from public databases. Data citations in the article text are distinct from normal bibliographical citations and should directly link to the database records from which the data can be accessed. In the main text, data citations are formatted as follows: "Data ref: Smith et al, 2001" or "Data ref: NCBI Sequence Read Archive PRJNA342805, 2017". In the Reference list, data citations must be labeled with "[DATASET]". A data reference must provide the database name, accession number/identifiers, and a resolvable link to the landing page from which the data can be accessed at the end of the reference. Further instructions are available at: <https://www.embopress.org/page/journal/14602075/authorguide#referencesformat>.

9. We request authors to consider both actual and perceived competing interests. Please review our policy (<https://www.embopress.org/page/journal/14602075/authorguide#conflictsofinterest>) and update your competing interests statement if necessary. Please name this section 'Disclosure and competing interests statement' and place it after the Acknowledgements section.

10. Please note that all corresponding authors are required to provide an ORCID ID upon submission of a revised manuscript (<https://orcid.org/>). Please find instructions on how to link your ORCID ID to your account in our manuscript tracking system in our Author guidelines (<https://www.embopress.org/page/journal/14602075/authorguide#authorshipguidelines>).

11. We use CRediT to specify the contributions of each author in the journal submission system. CRediT replaces the author

contribution section, which should be removed from the manuscript. Please use the free text box to provide more detailed descriptions. See also guide to authors:

<https://www.embopress.org/page/journal/14602075/authorguide#authorshipguidelines>.

13. We would also welcome the submission of cover suggestions or motifs to be used by our Graphics Illustrator in designing a cover.

14. Please use the link below to submit your revision:

Referee #1:

Cretu et al describe the single particle cryo EM structures of human ferlins (myoferlin and dysferlin) in the Ca²⁺ and lipid-bound state. These structural data are complemented with functional and MS data to support the conclusions of the two structures. Nanodiscs were used as lipid mimetics. In contrast, the determined structure does not resemble the currently proposed models. I am truly not an expert in the research of ferlins, but the data presented are solid. The overall resolution of the structures also permits the conclusions drawn by the authors. Equally important, proper control experiments have been performed. I also agree that these structures and the conclusions will serve as a general blueprint for the cellular function of human ferlins and recommend acceptance of the manuscript.

Referee #2:

This manuscript from investigates the conformational rearrangements occurring in two ferlin proteins, myoferlin and dysferlin, upon binding to membranes. Ferlins are multi-C2 domain proteins that mediate a wide range of Ca²⁺-dependent vesicle fusion processes. Here, the authors use cryoEM to determine the structures of human myoferlin and dysferlin in their Ca²⁺- and membrane-bound states as well as in cytosolic forms. They find that these proteins adopt a ring-like conformation where the C2C-C2D region is rigid and varies little across the observed functional states. Conversely, the C2B, C2F, and C2G domains are dynamic and undergo large scale conformations upon membrane binding. The authors use chemical crosslinking and mass spec to validate the observed architecture.

Overall, this is an excellent and well-written manuscript. The structural data is of high quality, the crosslinking and mass spec data is interesting. I have a few suggestions to solidify the conclusions and to improve the clarity of the presentation.

Is it possible that the limited size of a nanodisc membrane influences the overall arrangement of human myoferlin, such that a closed ring-like structure is favored? Ideally, a liposome bound structure would unambiguously address this question. However, I realize this might be technically challenging. As an alternative, would it be feasible to repeat the crosslinking and tandem mass spectrometry experiments on dysferlin bound to liposomes? Or is the partition constant too low to effectively separate the lipid bound molecules from the ones in solution? If direct experimental evidence is too difficult to obtain, then the authors could at least consider this point in the discussion.

The description of the myoferlin-membrane interface is confusing. The authors conclude that "membrane recognition by human myoferlin is achieved through both Ca²⁺-dependent and Ca²⁺-independent mechanisms", however neither the figures nor the text provide much clarity on this important point. Only Fig 2b shows the position of the C2C Ca²⁺ sites relative to the membrane, and even those do not appear to be so close to the membrane. Absent a structural comparison to a Ca²⁺ free structure, it is unclear how this region rearranges upon Ca²⁺ binding to support the authors' claims. Further, neither the C2C nor the C2D domains move much between lipid-bound and soluble states (Fig. 7), so it is unclear what these mechanisms could be. I suggest the authors add figures illustrating comparisons to Ca²⁺-free structures of relevant C2 domains -where available. Finally, I am a bit confused by the authors' claims that "the N-terminal C2B and the C-terminal C2F-C2G domains cycle between alternative conformations and, in response to Ca²⁺", as no Ca²⁺-free structures are reported.

I find the coloring scheme in Fig. 7 very confusing. Some domains change colors between panels (b) and (c), but some don't. The figure legend does not provide insights, so a reader is left to infer the meaning of the color changes based on intuition. Maybe simply having pale and solid colors indicate soluble and lipid bound structures would help. Additionally, I suggest the authors generate a morph movie between the lipid-bound and soluble forms of myoferlin to help visualize the movements of the various domains that occur upon binding to the membrane.

Reviewer #1

Cretu et al describe the single particle cryo-EM structures of human ferlins (myoferlin and dysferlin) in the Ca²⁺ and lipid-bound state. These structural data are complemented with functional and MS data to support the conclusions of the two structures. Nanodiscs were used as lipid mimetics. In contrast, the determined structure does not resemble the currently proposed models.

I am truly not an expert in the research of ferlins, but the data presented are solid. The overall resolution of the structures also permits the conclusions drawn by the authors. Equally important, proper control experiments have been performed. I also agree that these structures and the conclusions will serve as a general blueprint for the cellular function of human ferlins and recommend acceptance of the manuscript.

We would like to thank the reviewer for appreciating our work and supporting its publication.

Reviewer #2

This manuscript from investigates the conformational rearrangements occurring in two ferlin proteins, myoferlin and dysferlin, upon binding to membranes. Ferlins are multi-C2 domain proteins that mediate a wide range of Ca²⁺ -dependent vesicle fusion processes. Here, the authors use cryoEM to determine the structures of human myoferlin and dysferlin in their Ca²⁺- and membrane-bound states as well as in cytosolic forms. They find that these proteins adopt a ring-like conformation where the C2C-C2D region is rigid and varies little across the observed functional states. Conversely, the C2B, C2F, and C2G domains are dynamic and undergo large scale conformations upon membrane binding. The authors use chemical crosslinking and mass spec to validate the observed architecture.

Overall, this is an excellent and well-written manuscript. The structural data is of high quality, the crosslinking and mass spec data is interesting. I have a few suggestions to solidify the conclusions and to improve the clarity of the presentation.

We would like to thank the reviewer for appreciating the work and her/his advice that helped us to improve the manuscript.

Is it possible that the limited size of a nanodisc membrane influences the overall arrangement of human myoferlin, such that a closed ring-like structure is favored? Ideally, a liposome bound structure would unambiguously address this question. However, I realize this might be technically challenging. As an alternative, would it be feasible to repeat the crosslinking and tandem mass spectrometry experiments on dysferlin bound to liposomes? Or is the partition constant too low to effectively separate the lipid bound molecules from the ones in solution? If direct experimental evidence is too difficult to obtain, then the authors could at least consider this point in the discussion.

Thank you for this valuable comment. Indeed, obtaining high-resolution structural information on liposome-bound samples is challenging with very few successful examples of model samples (e.g., Melville et al., *Structure*, 2022, DOI: 10.1016/j.str.2021.08.001; Yao et al., *PNAS*, 2020, DOI: 10.1073/pnas.2009385117; Kovtun et al., *Science Advances*, 2020, DOI: 10.1126/sciadv.aba838), which we, nevertheless, hope to pursue in future experiments. Our choice of the membrane scaffold (MSP2N2) was based on the study by Cannon et al. (*J Struct Biol* 2023, DOI: 10.1016/j.jsb.2023.107989), in which the authors demonstrated that large pre-assembled MSP2N2 nanodiscs are suitable surrogates for single-particle cryo-EM of membrane-interacting samples and impose less conformational bias compared to other available nanodisc scaffolds. We, therefore, believe that nanodisc-induced artifacts in myoferlin's lipid-bound structure are unlikely, for several reasons:

(i). AlphaFold2 and AlphaFold3 predictions of myoferlin and dysferlin result in conformations reminiscent of both the experimentally observed lipid-bound and lipid-free myoferlin states (<https://deepmind.google/technologies/alphafold/alphafold-server/>).

(ii). Processing of a lipid-free otoferlin dataset (*unpublished data*, part of a large collaborative project including multiscale functional and morphological analysis of otoferlin mouse mutants the results of which are currently prepared for publication, Fig. R1-R3), reveals the presence of closed ring conformations (lipid-bound-like states) in the absence of nanodiscs (Fig. R3), suggesting that the closed ferlin state is generally sampled in solution, rather than being induced upon lipid bilayer binding (i.e., conformational selection). The otoferlin closed state, observed in both the lipid-free and nanodisc-bound datasets, is reminiscent of the closed state of myoferlin reported in this study. Moreover, lipid-bound otoferlin has a reduced nanodisc interface (involving only the C₂B and C₂F-C₂G domains), suggesting that the employed MSP2N2 nanodisc allows for alternative membrane binding modalities, rather than forcing a nonspecific, generally closed conformation of the ferlin ring.

(iii). All attempts at replication using various experimental setups (e.g., different lipid nanodisc compositions, salt and crosslinker concentrations) resulted in a generally similar lipid-bound closed state (e.g., Fig. EV2 and Appendix Fig. S2, S4, and S5).

In response to the reviewer's comment, we have now rephrased and provide discussion of this aspect:

Results:

“In our attempts to identify an optimal membrane system, we observed that the cytosolic region of myoferlin (residues 1-1997) formed stable complexes with large MSP2N2 (membrane scaffold protein 2 N2)-based lipid nanodiscs, comprising anionic phospholipids and chosen to accommodate all interacting domains (Cannon, Sarsam et al., 2023) (Fig 1D and Appendix Fig S1B-C).”

Discussion:

“Contrary to previous models (Dominguez et al., 2022, Woolger et al., 2017, Xu et al., 2011), the ferlin C₂ domains are not trivially organized as “beads on a string” or as in other, well-studied multi-C₂ domain factors (Schauder et al., 2014, Shin et al., 2005). Instead, the conserved structural motifs pack uniquely in 3D and form state-defining interfaces, bridging both neighbouring and sequence-distant domains.”

and

“Future functional and structural studies are needed to probe the validity of the proposed mechanisms, beyond simple nanodisc membranes, particularly within the context of pathway-defining, higher-order ferlin complexes (Grushin et al., 2019, Stepien et al., 2022, Zhou et al., 2015, Zhou et al., 2017). These studies will likely entail the use of alternative imaging modalities (cryo-electron tomography and super-resolution imaging) and computational approaches (molecular dynamics simulations), applied to both surrogate membrane systems (liposomes) and in vivo triggered vesicle fusion (Chakrabarti et al., 2022, Held et al., 2024, Imig et al., 2020, Kovtun et al., 2020, Rizo et al., 2022, Wang et al., 2024). Such complementary analyses will help exclude improbable non-physiological myoferlin conformations induced by nanodiscs and further elucidate the roles of additional ferlin molecules and ferlin-SNAREs interactions in the vesicle-membrane fusion cycle.”

The description of the myoferlin-membrane interface is confusing. The authors conclude that "membrane recognition by human myoferlin is achieved through both Ca²⁺-dependent and Ca²⁺-independent mechanisms", however neither the figures nor the text provide much clarity on this important point. Only Fig 2b shows the position of the C₂C Ca²⁺ sites relative to the membrane, and even those do not appear to be so close to the membrane. Absent a structural comparison to a Ca²⁺ free structure, it is unclear how this region rearranges upon Ca²⁺ binding to support the authors' claims. Further, neither the C₂C nor the C₂D domains move much between lipid-bound and soluble states (Fig. 7), so it is unclear what these mechanisms could be. I suggest the authors add figures illustrating comparisons to Ca²⁺-free structures of relevant C₂ domains -where available. Finally, I am a bit confused by the authors' claims that "the N-terminal C₂B and the C-terminal C₂F-C₂G domains cycle between alternative conformations and, in response to Ca²⁺", as no Ca²⁺-free structures are reported.

Thank you for the important comment. We have now restructured this section, along with the related figures (Fig. 4 and Fig. 5), to improve clarity. Solving Ca²⁺-free and liposome-bound ferlin structures (myoferlin and otoferlin) remains an important objective for future studies; however, we believe that, on technical grounds, this lies beyond the scope of the present work.

When discussing the Ca²⁺-dependency of membrane binding, we refer to direct, phospholipid-specific contacts via the coordinated Ca²⁺ ions (by the acidic residues of the top L1 and L3 loops). In the cases of C₂C and C₂F, we consistently observe (e.g., Fig EV1C and Fig EV1M) that phospholipid headgroups (most likely phosphatidylserine), originating from the nanodisc bilayer, form coordination bonds with 1-2 bound Ca²⁺ ions (Ca²⁺-dependent interaction). However, this membrane interaction mechanism does not apply to all of myoferlin's C₂ domains. One interesting insight from our structures is that not all C₂ domains that bind Ca²⁺ interact with the nanodisc surface (e.g., C₂D), and not all C₂ domain-lipid interactions require Ca²⁺ binding (e.g., C₂B). Moreover, at least one tertiary interface of myoferlin – the C₂D-C₂E interface – involves the Ca²⁺-binding loops of C₂D, suggesting that Ca²⁺ not only promotes membrane binding (by C₂C, C₂F, and C₂G) but could also modulate the conformational

transition toward the closed state. These findings highlight the complexity of ferlin-membrane interactions, which could not have been anticipated from studies of individual C₂ domains.

In response to the reviewer's comment, we have also updated Fig. 4 and Fig. 6 to better convey our findings and analyses.

I find the coloring scheme in Fig. 7 very confusing. Some domains change colors between panels (b) and (c), but some don't. The figure legend does not provide insights, so a reader is left to infer the meaning of the color changes based on intuition. Maybe simply having pale and solid colors indicate soluble and lipid bound structures would help. Additionally, I suggest the authors generate a morph movie between the lipid-bound and soluble forms of myoferlin to help visualize the movements of the various domains that occur upon binding to the membrane.

We would like to thank the reviewer for their suggestions, which we have implemented in the revised Fig. 7. We have also included two movies (Movie EV4 and Movie EV5), including a morph movie (Movie EV5), to better visualize the suggested conformational transition.

Dear Tobias,

Congratulations on an excellent manuscript! I am very pleased to inform you that it has been accepted for publication in The EMBO Journal. Thank you very much for addressing the referees' concerns and the editorial and formatting requests.

If you have any questions, please do not hesitate to contact the Editorial Office. Thank you for your contribution to The EMBO Journal. Working with you has been a pleasure.

Best wishes,

Ioannis
